# Dualformer: Controllable Fast and Slow Thinking by Learning with Randomized Reasoning Traces

**DiJia Su   Sainbayar Sukhbaatar   Michael Rabbat   Yuandong Tian   Qinqing Zheng**
Meta AI

## Abstract

In cognition theory, human thinking is governed by two systems: the fast and intuitive System 1 and the slower but more deliberative System 2. Analogously, Large Language Models (LLMs) can operate in two reasoning modes: outputting only the solutions (*fast mode*) or both the reasoning chain and the final solution (*slow mode*). We present Dualformer, a single Transformer model that seamlessly integrates both the fast and slow reasoning modes by training on randomized reasoning traces, where different parts of the traces are strategically dropped during training. At inference time, Dualformer can be easily configured to execute in either fast or slow mode, or automatically decide which mode to engage (*auto mode*). It outperforms baselines in both performance and computational efficiency across all three modes: **(1)** in slow mode, Dualformer achieves $97.6\%$ optimal rate on unseen $30 \times 30$ maze tasks, surpassing the Searchformer baseline ($93.3\%$) trained on data with complete reasoning traces, with $45.5\%$ fewer reasoning steps; **(2)** in fast mode, Dualformer achieves $80\%$ optimal rate, significantly outperforming the Solution-Only model trained on solution-only data, which has an optimal rate of only $30\%$; **(3)** in auto mode, Dualformer achieves $96.6\%$ optimal rate with $59.9\%$ fewer steps than Searchformer. Moreover, Dualformer produces more diverse reasoning traces than Searchformer. For math reasoning problems, our techniques have also achieved improved performance with LLM fine-tuning, demonstrating its generalization beyond task-specific models. We open source our code at https://github.com/facebookresearch/dualformer.

## 1 Introduction

Transformers (Vaswani et al., 2017), the sequence modeling tool that serves as the cornerstone of foundation models in various domains including Large Language models (LLMs) (Dosovitskiy, 2020; Baevski et al., 2020; Radford et al., 2021; Touvron et al., 2021; Hsu et al., 2021; Touvron et al., 2023; Dubey et al., 2024), have been widely used in many works to approach reasoning and planning problems, see e.g., Zhou et al. (2022); Kojima et al. (2022); Pallagani et al. (2022); Valmeekam et al. (2023a); Chen et al. (2024); Gundawar et al. (2024); Wang & Zhou (2024). Specifically, we can categorize the reasoning modes of Transformers into *fast* and *slow*. In fast mode, a Transformer will output a final solution without any reasoning steps, whereas the intermediate steps of thinking, such as a search trace for finding a short path, will be generated along with the plan in slow mode.

The two inference modes share a lot of similarities with the two thinking systems inherent in us (Wason & Evans, 1974; Kahneman, 2017): an automatic and unconscious System 1 and a controlled and conscious System 2. More importantly, they come with analogous pros and cons. As discussed in previous works (Wei et al., 2022; Valmeekam et al., 2023b; Lehnert et al., 2024; Gandhi et al., 2024; Saha et al., 2024), Transformer models operating in fast mode have a lower computational cost and allow for a quicker response, yet they fall short in accuracy and optimality compared with slow mode models (see Figure B.1 for a concrete example). This raises an interesting question:

> *Can we integrate both the fast and the slow modes into Transformer-based reasoning models, similar to how humans possess two distinct thinking systems, and let them complement each other?*

Multiple approaches have been proposed. A popular paradigm is to start with a pure System 2, and then fine-tune to make it more efficient like System 1, e.g., to distill System 2 output into System 1 (Yu et al., 2024; Deng et al., 2024), or to improve the reasoning efficiency of existing system 2 by distillation (Wang et al., 2023; Shridhar et al., 2023) or bootstrapping from symbolic systems (e.g., `Searchformer` (Lehnert et al., 2024), Stream of Search (Gandhi et al., 2024)). However, in these cases, further fine-tuning is needed, which is computationally expensive, and it is non-trivial to adapt the resulting system to be more like System 1 or System 2 on the fly. To address this problem, Saha et al. (2024) design an explicit meta-controller to switch between two different systems.

In this work, we demonstrate a surprising finding: *a simple data recipe suffices to achieve on-the-fly System 1 and System 2 configuration in solving reasoning tasks*. The resulting model, `Dualformer`, can be easily configured to execute in either fast or slow mode during inference, and determines which mode to use by itself if not specified. More specifically, to imitate the System 2 reasoning process, our Transformer is trained on data that contains both the reasoning trace and the final solution. Leveraging the structure of reasoning steps, we design specific trace dropping strategies such that the resulting traces resemble the shortcuts taken by System 1 in the thinking process. In the extreme case, we drop the entire trace and encourage the Transformer to output a final solution directly, bypassing all the intermediate steps. We randomly choose those structured trace-dropping strategies at training time. See Section 3 for the details.

We first apply our framework to train an encoder-decoder Transformer model to solve pathfinding problems, where the trace is generated by the A* search algorithm. We consider two domains: the Maze navigation and the Sokoban game as in Lehnert et al. (2024), where we use the same tokenization scheme. Interestingly, we have found that these problems are challenging for state-of-the-art LLMs like o1-preview and o1-mini, where the output path often breaks into the walls (See Appendix I for an example). In each reasoning mode, `Dualformer` outperforms the established baselines, achieving stronger results in both the solved rate and optimal rate. Moreover, `Dualformer` significantly enhances the diversity of the generated plans by identifying a greater variety of unique paths that reach the goal. Notably, `Dualformer` is also efficient even when working in slow mode, generating much shorter reasoning traces than the baseline model. Next, we apply our framework to fine-tune LLMs for answering math questions. Following Yu et al. (2023), the training examples are taken from the `MATH` dataset (Hendrycks et al., 2021) where answers are rewritten by a Llama-3.1-70B-Instruct model to include detailed intermediate steps. Likewise, the obtained LLMs demonstrate enhanced efficacy and efficiency.

## 2 RELATED WORK

**Learning to Plan and Reason** Tremendous efforts have been made to enhance the capability of Transformer-based models to plan and reason over a long horizon. Two main types of approaches have been developed. The first type leverages existing LLMs. For instance, researchers taught LLMs to call external existing symbolic solvers, such as those found in Ahn et al. (2022); Besta et al. (2024); Sel et al. (2023); He-Yueya et al. (2023); Liu et al. (2023); Silver et al. (2024). Pallagani et al. (2022; 2024) investigate the fine-tuning of LLMs to use a symbolic solver, (Schick et al.; Hao et al., 2024) fine-tune LLMs to use external tools, and (Hao et al.) propose to fine-tune LLMs to do reasoning and planning within a world model. Among them, several works try to integrate System-1 and System-2 thinking into LLM reasoning. Weston & Sukhbaatar (2023) proposed the System 2 Attention (S2A), a more deliberate attention mechanism aimed at improving the reasoning capabilities of large language models by reducing their susceptibility to spurious correlations in the context. Yu et al. (2024) distill System 2 output into a System 1 model, which aims to compile higher-quality outputs from System 2 techniques back into LLM generations without intermediate reasoning token sequences.

The second type aims to train Transformers from scratch to plan and reason independently (Lehnert et al., 2024; Saha et al., 2024; Gandhi et al., 2024), using task-specific language. Both Lehnert et al. (2024) and Gandhi et al. (2024) teach LLMs to search by representing the search process in language, Lehnert et al. (2024) developed `Searchformer` that is trained to mimic the A* search process for pathfinding problems. Gandhi et al. (2024) applied their model to the Countdown game. In the

concurrent work Saha et al. (2024), the authors trained two separate models and manage them by an external meta-controller. There, one model generates rapid traceless response, while another one responds slower but with trace. Similarly, (Lin et al., 2023) leverages multiple models to implement slow and fast thinking using an agent style workflow. Our research is closely related to the above work, but has some key differences. Unlike Lehnert et al. (2024); Gandhi et al. (2024) which do not modify the reasoning trace in the training data, our approach involves randomizing the reasoning trace. Unlike Saha et al. (2024); Lin et al. (2023), we do not use any explicit controller nor use two networks for each mode. Instead, we integrate both the fast and the slow mode functionalities into a single model.

**Synthetic Data Generation using LLM** Large language models (LLMs) have been utilized for synthetic data generation in various domains. For instance, Wei et al. (2021); Longpre et al. (2023); Taori et al. (2023) introduced a synthetic instruction dataset by sampling from diverse templates containing natural language instructions that outline specific tasks. This approach has also been applied to the visual domain (Liu et al., 2024b;a; Zhu et al., 2023; Brooks et al., 2023; Peng et al., 2023). To improve the performance of LLM to answer math questions, Yu et al. (2023) developed a method to rewrite, verify, and augment the original MATH dataset (Hendrycks et al., 2021) using specialized prompts. Similar methodologies have been further explored in other studies, including those by Yuan et al. (2023); Luo et al. (2023); Lee et al. (2023); Yue et al. (2023); Tong et al. (2025).

## 3   STRUCTURED TRACE DROPPING AND RANDOMIZED TRAINING

Our work builds upon the `Searchformer` work of Lehnert et al. (2024). To perform planning, we train a Transformer to model a sequence of tokens that sequentially represents the planning task (*prompt*), the computation of A* algorithm (*trace*), and the optimal path (*solution*) derived. See Appendix A to the details of our tokenization scheme and example input for a Maze navigation task.

`Searchformer` has proven efficacy in addressing a variety of complex decision-making tasks. However, it still suffers from two important limitations. Firstly, the model only operates in slow mode and outputs lengthy reasoning chains, which significantly increases the inference time. While this can be reduced by bootstrapping (Lehnert et al., 2024), an iterative refining technique that consists of cycles of rollouts followed by fine-tuning, such a procedure incurs significant extra demand on computational resources. Secondly, `Searchformer` struggles to generate diverse solutions, where identical rollouts are frequently sampled[1]. For example, across 1000 $30 \times 30$ maze problems we have tested, `Searchformer`'s reasoning chain contains more than 1500 tokens on average, and can only find 7.6 unique feasible paths out of 64 responses (see Section 4.1.2).

To address these challenges, we propose a training framework that utilizes randomized reasoning traces. Our approach is inspired by two lines of work. First, we have noticed that even though `Searchformer` is trained on complete A* search traces, it generates shorter traces that are sketching the search process. Second, research has shown that humans often rely on shortcuts and patterns when making decisions, a concept known as System 1 thinking (Kahneman, 2017). These observations, combined with the success of dropout technique (Hinton, 2012; Srivastava et al., 2014) that randomly drop units from the neural network during training, motivated us to explore the use of randomized reasoning traces in our framework, and we aim to simplify the A* search trace by exploiting its structured elements and selectively dropping certain parts for each training example.

As shown in Figure 3.1, the A* search trace contains both the `create` and the `close` clauses, and each clause includes the node's coordinates and its (estimated) cost to reach the start and the goal locations. To derive `Dualformer`, we exploit the structure of the search trace and drop certain parts of it for each training example. There are three natural types of dropping:

- D1: drop a `close` clause

- D2: drop the cost tokens in a clause

- D3: drop a `create` clause

---

[1]We note that the lack of diversity is not due to the limitation of training data. For all our experiments, the search traces in the training datasets are generated by non-deterministic A* algorithm that randomly breaks cost ties.

**Figure 3.1:** An illustration of the structured trace dropping strategies, which consists of four levels. Each level employs a progressively more aggressive dropping approach than the previous one. For an introduction of trace format, see Appendix A.

Based on them, we develop four levels of dropping strategies, each building upon the previous one. Specifically,

- `Level 1` strategy eliminates all the `close` clauses from a search trace.

- `Level 2` strategy goes further by additionally dropping all the cost tokens.

- `Level 3` strategy is even more aggressive. It further randomly drops $30\%$ of the `create` clauses.

- `Level 4` strategy drops the entire trace.

Figure 3.1 illustrates our strategies using the previously mentioned Maze task. Intuitively, the `Level 1` dropping instructs `Dualformer` to effectively bypass the close-set computation of A* search, the `Level 2` dropping promotes `Dualformer` to bypass both the close-set and the cost computation. The `Level 3` and `Level 4` dropping encourage `Dualformer` to omit certain or all of the search steps. As we will show in Section 4, these strategies effectively guide `Dualformer` in learning a more concise and efficient search and reasoning process.

To promote diversity in the training data, we do not perform dropping as a data preprocessing step. Instead, at training time, for each training example within a batch, we randomly sample the dropping strategy from a categorical distribution $\text{Cat}(p_0, p_1, p_2, p_3, p_4)$, where $p_1, \ldots, p_4$ are the probabilities of performing `Level 1-4` droppings, and $p_0$ is the probability of maintaining a complete trace. This training framework enables `Dualformer` to learn from multiple reduced traces even for a single training example, as the same example might appear in multiple batches.

**Comparison with Token Masking** Curious readers might already be wondering that whether our training framework resembles the token masking techniques used by famous LLMs including BERT (Devlin, 2018; Liu, 2019; Song, 2019; Gauthier & Levy, 2019; Sinha et al., 2021; Kitouni et al., 2024). However, there are significant differences that distinguish our approach from those masking techniques. First, standard masking techniques usually mask the input tokens of a sequence uniformly in random. In contrast, our dropping strategies only apply to the search trace. Second, while masked LLMs generally employ bidirectional attention layers and predict the masked tokens, `Dualformer` uses causal attention layers, and our training objective solely focuses on the next token prediction with the overall goal of improving its reasoning and planning capability. Computationally, our training procedure is also more efficient. Dropping tokens shortens the input sequence and saves computation. For example, it takes 30 hours to train `Dualformer` for the $30 \times 30$ maze task on 8 Tesla V100 32GB GPUs, while it requires 36 hours if we use the full reasoning trace. We defer the training details to Section 4 and Appendix C.

## 3.1 CONTROLLABLE GENERATION

One appealing property of `Dualformer` is that it can be easily prompted to operate in either fast or slow generation mode at inference time. The control mechanism is extremely simple: we append `bos` and a *control* token to the standard prompt (which includes environment and task description), where the control token is either `plan` or `create`. If we use `plan`, `Dualformer` will operate in fast mode and directly output the plan, bypassing the reasoning steps. On the other hand, if we inject `create` after `bos`, `Dualformer` will work in slow mode and generate both the reasoning trace and the final plan. See Appendix D for concrete examples. If we only use the standard prompt, `Dualformer` will mimic the dual process of human decision making—depending on the situation, it generates either types of responses, which correspond to System 1 and System 2 reasoning.

## 4 EXPERIMENTS

Our experiments are designed to answer the following questions:

1. Does `Dualformer` outperform corresponding baselines in fast, slow and auto mode? Does it generate more diverse plans?
2. In slow and auto model, does `Dualformer` lead to faster reasoning, i.e., output a shorter trace?
3. Does the structured trace dropping technique generalize to LLMs trained on natural language datasets?

We answer questions 1 and 2 in Section 4.1, where we train Transformers to solve Maze navigation tasks and Sokoban games, similar to `Searchformer` (Lehnert et al., 2024). To answer questions 3, we fine-tune LLama-3.1-8B and Mistral-7B models to solve math problems in Section 4.2.

### 4.1 NAVIGATION TASKS: MAZE & SOKOBAN

Following Lehnert et al. (2024), we consider the Maze and the Sokoban tasks and use the same dataset. Both datasets contain $10^5$ training examples with complete A* search trace. The A* implementation is non-deterministic where it breaks cost ties randomly and randomizes the order in which child nodes are expanded. The size of Maze varies from $15 \times 15$ to $30 \times 30$. For all the Maze tasks, we randomly generate $30\% - 50\%$ percentage of wall cells as obstacles, and randomly sample the goal and start locations. The Sokoban map is of size $7 \times 7$ with randomly placed two docks, two boxes, and the worker location. We also randomly add two additional wall cells to the interior of the map. For the detailed map generation procedure and example figures, we refer the readers to Appendix C.1.

We first demonstrate that `Dualformer` can be explicitly controlled to operate in either fast or slow mode in Section 3.1. It will only output the final plan in fast mode, while a reasoning trace will be generated in slow mode. In Section 4.1.1-4.1.2, we compare `Dualformer` with the corresponding baselines in each mode, respectively. A variety of metrics, as listed below, are used to systematically evaluate the performance, including the correctness, optimality and diversity of generated plans, length of the reasoning trace, etc. Last, we ablate our design choices in Section 4.1.5.

**Hyperparameters** We instantiate `Dualformer` using the same encoder-decoder architecture as in Lehnert et al. (2024). The encoder is an adaptation of the T5 architecture (Raffel et al., 2020) with rotary embeddings, while the decoder is a GPT style architecture. We employ a model size of 15M and 46M for the Maze and the Sokoban environments, respectively. All models are trained on 100k training examples. We trained the model for $4 \times 10^5$ iterations for the Maze environment and $8 \times 10^5$ iterations for the Sokoban environment. We defer the details of model architectures and the other hyperparameters to Appendix C.

We train `Dualformer` using the structured trace dropping strategies described in Section 3. For choosing the dropping strategies for each training example, we sweep 3 sets of probabilities (1) $\{p_0 = 0.45, p_1 = p_2 = p_3 = 1/6, p_4 = 0.05\}$, (2) $\{p_0 = 0.8, p_1 = p_2 = p_3 = p_4 = 0.05\}$, (3)$\{p_0 = 0.7, p_1 = 0.05, p_2 = p_3 = 0.1, p_4 = 0.05\}$ and choose the set that yields the lowest validation error. The final choices are (1) for Maze and (3) for Sokoban.

**Baselines** For the fast mode, our baseline is the *Solution-Only* model, which uses the same architecture as `Dualformer` but trained on sequence data that only include the optimal final solution, without any reasoning trace. The slow mode baseline is the *Complete-Trace* model trained on data with complete A* search traces. It is referred to as the *search augmented* model in Lehnert et al. (2024), and is also the base `Searchformer` model without search dynamics bootstrapping. We will also compare `Dualformer` with bootstrapped models in Section 4.1.2. All these models have the same amount of parameters, 15M for Maze problems and 46M for Sokoban problems.

**Metrics** We evaluate whether a model generates correct and optimal plans using two metrics: *1-Solved-64* and *1-Optimal-64*. Namely, for each evaluation task (e.g. one maze or one Sokoban game), we randomly sample 64 responses from a trained model. Each response is parsed and evaluated regardless of the generated trace part. If any of the 64 plans is correct, i.e. is feasible and reaches the goal location, this task is labelled as success for the *1-Solved-64* metric. If any of the 64 plans is optimal, this task is labelled as success for the *1-Optimal-64* metric. We repeat such process for 1000 unseen evaluation tasks and report the average success rate. To investigate the robustness

of each method, we also report metrics *3-Solved-64* and *3-Optimal-64* where a task is labelled as success if at least 3 plans are correct or optimal. Additionally, we consider the *Success Weighted by Cost (SWC)* (Wu et al., 2019) that measures the quality of the resulting plans in terms of their costs, aggregated over individual responses. More precisely: $\text{SWC} = \frac{1}{64} \sum_{i=1}^{64} \mathbb{I}(\text{plan } i \text{ is correct}) \cdot \frac{c^*}{c_i}$, where $\mathbb{I}$ is the indicator function, $c^*$ is the cost of an optimal plan, and $c_i$ is the cost of the $i$-th plan. Clearly, the higher the SWC score is, the more optimal the resulting plans are. If all the generated plans are optimal, the SWC score reaches its maximum value 1. Last, to quantify the diversity of the generated plans, we check the number of unique correct plans out of the 64 responses for each task, and report the average number across 1000 evaluation tasks.

### 4.1.1 FAST MODE

Table 4.1 reports the performance of `Dualformer` and the baseline Solution-Only model on Maze and Sokoban tasks, respectively. In terms of generating correct and optimal plans, `Dualformer` significantly outperforms the baseline in both 1-Solved-64 and 1-Optimal-64 criteria. It also notably surpasses the baseline in terms of the 3-Solved-64 and 3-Optimal-64 rates, which demonstrates the robustness of `Dualformer` in plan generation. In particular, the performance gap increases as the task difficulty increases. For the largest $30 \times 30$ Maze, the 1-Optimal-64 rate of `Dualformer` is $2.8\times$ that of Solution-Only model, and the 3-Optimal-64 rate is $2.97\times$. `Dualformer` also achieves much higher SWC score than the baseline, which is above $0.9$ for every environment. This demonstrates that each individual plan `Dualformer` generated is of high quality, whose cost is very close to the optimal plan.

`Dualformer` also consistently generates more diverse plans for all the considered problems. In Appendix E, we show one Maze example and plot the unique correct plans generated by `Dualformer` and the baseline model. One interesting observation is that the diversity score of `Dualformer`, i.e., the average number of unique correct plans out of 64 responses, increases as the Maze size goes up. Intuitively, as the Maze becomes larger, there are more possible routes to reach a single goal location. This suggests that `Dualformer` learns the Maze structure, while the Solution-Only model is potentially memorizing the optimal plans as its diversity score is close to 1 for all the Maze sizes.

### 4.1.2 SLOW MODE

Table 4.2 reports the results when `Dualformer` operates in slow mode. The corresponding baseline is the Complete-Trace model, which uses the same architecture and is trained on data with complete A* search traces. In addition to the metrics reported before, we report the average length of reasoning traces across the 64 responses, aggregated over all the 1000 evaluation tasks. The results show that `Dualformer` achieves both enhanced planning power and reasoning speed. It outperforms the Complete-Trace model for all the correctness and optimality metrics: solved rates, optimal rates, and SWC. Moreover, the reasoning trace yielded by `Dualformer` is notably shorter than the baseline model. On average, `Dualformer` reduces the trace length by 49.4% across the five tasks. As before, `Dualformer` also generates more diverse plans compared with the baseline. We refer the readers to Appendix E for concrete examples.

**Comparison with Search Dynamics Bootstrapping** The Complete-Trace model is the base `Searchformer` model in Lehnert et al. (2024), which has also proposed a search dynamics bootstrapping method to enhance its performance on the Sokoban task, similar to those in Anthony et al. (2017); Zelikman et al. (2022). After training the `Searchformer` model, we fine-tune it on a newly created self-bootstrapped dataset. For each Sokoban game in the original dataset, we generate 32 responses and include the shortest optimal response into the new dataset. We can repeat this process multiple times. In this way, the `Searchformer` learns to generate shorter responses. Table 4.4 compares `Dualformer` with `Searchformer` models fine-tuned up to 3 steps. `Dualformer` is comparable or better than bootstrapped models in most of the metrics, while only using fewer than 45.1% reasoning steps. We note that each bootstrapping step requires rollouting $3.2 \times 10^6$ total responses and extra fine-tuning of $10^4$ iterations, This means, including the $8 \times 10^5$ pretraining iterations, `Searchformer` step 3 requires a total of $8.3 \times 10^5$ training iterations and $9.6 \times 10^6$ rollouts, which is expensive in computation. In comparison, `Dualformer` only needs a single training stage that consists of $8 \times 10^5$ iterations, with no additional rollout requirements.

| | Method | 1-Optimal-64 / 3-Optimal-64 | 1-Solved-64 / 3-Solved-64 | SWC | Diversity |
|---|---|---|---|---|---|
| Maze 15x15 | Dualformer (fast) | 91.8 / 87.6 | 97.1 / 94.8 | 0.960 | 9.05 |
| | Solution-Only | 72.0 / 68.9 | 82.7 / 80.1 | 0.610 | 1.52 |
| Maze 20x20 | Dualformer (fast) | 90.9 / 84.0 | 97.0 / 94.0 | 0.960 | 17.27 |
| | Solution-Only | 56.3 / 52.0 | 71.9 / 67.5 | 0.690 | 1.52 |
| Maze 25x25 | Dualformer (fast) | 83.9 / 72.9 | 95.5 / 90.6 | 0.940 | 21.23 |
| | Solution-Only | 39.7 / 34.7 | 60.3 / 55.4 | 0.570 | 1.9 |
| Maze 30x30 | Dualformer (fast) | 80.0 / 66.0 | 91.8 / 85.7 | 0.906 | 18.23 |
| | Solution-Only | 30.0 / 26.0 | 54.1 / 47.8 | 0.500 | 1.86 |
| Sokoban | Dualformer (fast) | 97.3 / 94.4 | 94.8 / 90.0 | 0.970 | 4.92 |
| | Solution-Only | 86.8 / 83.4 | 92.8 / 90.0 | 0.919 | 1.24 |

**Table 4.1:** Evaluation performance of fast mode `Dualformer`. The baseline model is the same architecture trained on the solution only data.

| | Method | Avg Trace Length | 1-Optimal-64 / 3-Optimal-64 | 1-Solved-64 / 3-Solved-64 | SWC | Diversity |
|---|---|---|---|---|---|---|
| Maze 15 x 15 | Dualformer (slow) | 278 | 99.6 / 99.2 | 99.9 / 99.9 | 0.999 | 12.54 |
| | Complete-Trace | 495 | 94.6 / 90.1 | 96.7 / 93.0 | 0.964 | 7.60 |
| Maze 20 x 20 | Dualformer (slow) | 439 | 98.9 / 97.8 | 99.9 / 99.7 | 0.998 | 18.86 |
| | Complete-Trace | 851 | 98.3 / 95.5 | 98.8 / 93.00 | 0.987 | 14.53 |
| Maze 25 x 25 | Dualformer (slow) | 589 | 99.9 / 97.2 | 99.7 / 99.3 | 0.997 | 25.05 |
| | Complete-Trace | 1208 | 95.2 / 85.7 | 97.0 / 90.4 | 0.968 | 18.85 |
| Maze 30 x 30 | Dualformer (slow) | 854 | 97.6 / 93.2 | 99.5 / 98.2 | 0.993 | 25.77 |
| | Complete-Trace | 1538 | 93.3 / 82.4 | 95.9 / 88.1 | 0.964 | 7.60 |
| Sokoban | Dualformer (slow) | 1482 | 94.5 / 87.6 | 97.4 / 94.1 | 0.970 | 4.66 |
| | Complete-Trace | 3600 | 92.9 / 84.4 | 94.7 / 89.0 | 0.944 | 2.91 |

**Table 4.2:** Evaluation performance of slow mode `Dualformer`. The baseline model is the same architecture trained on the complete-trace data (`Searchformer`).

| | Method | Avg Trace Length | 1-Optimal-64 / 3-Optimal-64 | 1-Solved-64 / 3-Solved-64 | SWC | Diversity |
|---|---|---|---|---|---|---|
| Maze 15 x 15 | Dualformer (auto) | 222 | 99.7 / 99.4 | 99.9 / 99.8 | 0.999 | 12.52 |
| | Complete-Trace | 495 | 94.6 / 90.1 | 96.7 / 93.0 | 0.964 | 7.60 |
| | Solution-Only | - | 72.0 / 68.9 | 82.7 / 80.1 | 0.610 | 1.52 |
| Maze 20 x 20 | Dualformer (auto) | 351 | 99.5 / 98.6 | 99.9 / 99.3 | 0.997 | 20.28 |
| | Complete-Trace | 851 | 98.3 / 95.5 | 98.8 / 93.0 | 0.987 | 14.53 |
| | Solution-Only | - | 56.3 / 52.0 | 71.9 / 67.5 | 0.690 | 1.52 |
| Maze 25 x 25 | Dualformer (auto) | 427 | 98.6 / 96.9 | 99.8 / 99.0 | 0.998 | 24.81 |
| | Complete-Trace | 1208 | 95.2 / 85.7 | 97.0 / 90.4 | 0.968 | 18.85 |
| | Solution-Only | - | 39.7 / 34.7 | 60.3 / 55.4 | 0.570 | 1.9 |
| Maze 30 x 30 | Dualformer (auto) | 617 | 96.6 / 92.1 | 98.4 / 97.7 | 0.989 | 24.42 |
| | Complete-Trace | 1538 | 93.3 / 82.4 | 95.9 / 88.1 | 0.964 | 7.60 |
| | Solution-Only | - | 30.0 / 26.0 | 54.1 / 47.8 | 0.500 | 1.86 |
| Sokoban | Dualformer (auto) | 494 | 94.0 / 90.0 | 97.4 / 94.7 | 0.979 | 4.97 |
| | Complete-Trace | 3600 | 92.9 / 84.4 | 94.7 / 89.0 | 0.944 | 2.91 |
| | Solution-Only | - | 86.8 / 83.4 | 92.8 / 90.0 | 0.919 | 1.24 |

**Table 4.3:** Evaluation performance of auto mode `Dualformer`. The baselines are the Solution-Only and Complete-Trace (`Searchformer`) models.

| Method | Avg Trace Length | 1-Optimal-64 / 3-Optimal-64 | 1-Solved-64 / 3-Solved-64 | SWC | Diversity |
|---|---|---|---|---|---|
| Dualformer (slow) | 1482 | 94.5 / 87.6 | 97.4 / 94.1 | 0.970 | 4.66 |
| Searchformer Step 1 | 3785 | 94.4 / 91.2 | 95.9 / 92.4 | 0.957 | 2.19 |
| Searchformer Step 2 | 3507 | 94.9 / 91.5 | 96.7 / 92.9 | 0.965 | 2.27 |
| Searchformer Step 3 | 3283 | 94.5 / 91.4 | 96.6 / 94.4 | 0.964 | 2.48 |

**Table 4.4:** Performance for the Sokoban game. `Searchformer` is fine-tuned up to 3 steps using the search dynamics bootstrapping method.

### 4.1.3 AUTO MODE: DUAL PROCESS

Instead of controlling the inference mode of `Dualformer` by injecting a control token after `bos`, we can also sample from it directly, allowing it to freely determine the mode of operation, similar to the dual process of human decision making. We call this *auto mode* for `Dualformer`. Table 4.3

reported the results. The *auto mode* `Dualformer` also outperforms both the Complete-Trace and Solution-Only model for all the tasks we consider.

An interesting question to ask is whether `Dualformer` can automatically adjust its operation mode in response to problem difficulty. To investigate this, we generated 64 auto-mode responses for 1000 unseen mazes with varying wall densities (between 0.3 and 0.5). We then analyzed the percentage of slow-mode paths among all feasible solutions. The results are plotted in Figure 4.1. As the wall density increases, indicating a (likely) more challenging maze, the proportion of slow-mode paths also increases. This could be due to two factors: (1) the inherent need for slower thinking to solve more difficult problems, and (2) `Dualformer` is selecting the slow mode more frequently. Similarly, we observe that as the maze size increases, where the problem becomes harder, `Dualformer` consistently employs more slow thinking.

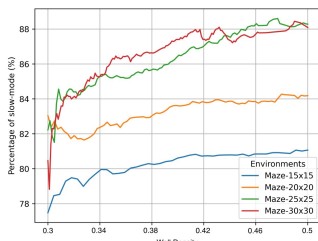

**Figure 4.1:** The percentage of slow mode paths (among all the feasible paths it generates) that `Dualformer` (auto mode) activates in. As the wall density and maze size, `Dualformer` engages in slow thinking more often.

### 4.1.4 GENERALIZATION PERFORMANCE

For all the experiments we have presented so far, we test on mazes of the same size and wall density. It is intriguing to inspect the OOD generalization performance of `Dualformer`. In this section, we consider a `Dualformer` trained on mazes of size $20 \times 20$, where wall density is uniform randomly sampled between $0.3$ and $0.5$. At test time, we vary the wall density from $0.1$ to $0.6$. Table 4.5 presents the slow mode results. We test 50 unseen examples for each case. As the wall density increases, we expect the maze to become more challenging and the prompt also becomes longer. Therefore, we can see for in-distribution test cases (wall density 0.3, 0.4, 0.5), `Dualformer` (slow mode) achieves approximately 100% optimal rate. Yet when the wall density increases to 0.6, which is OOD, the performance drops. Interestingly, `Dualformer` (slow mode) does not solve a single maze for lower wall densities. Our intuition is that the prompt becomes too short for `Dualformer` to generalize. In addition to varying wall densities, we also check the slow mode performance of on rectangular mazes (e.g., height=20, width=10). As before, our model is trained on $20 \times 20$ mazes. Table 4.6 shows the results. Surprisingly, `Dualformer` generalizes for the cases we consider here.

| Wall Density | 1-Optimal-64 | 1-Solved-64 |
|---|---|---|
| 0.1 (OOD) | 0.0 | 0.0 |
| 0.2 (OOD) | 0.0 | 0.0 |
| 0.3 | 97 | 100 |
| 0.4 | 100 | 100 |
| 0.5 | 100 | 100 |
| 0.6 (OOD) | 68 | 72 |

**Table 4.5:** `Dualformer` (slow mode, trained on $20 \times 20$ mazes) achieves nearly-perfect optimal rate when the wall density is in distribution, while the performance drops for OOD values.

| Maze Size | 1-Optimal-64 | 1-Solved-64 |
|---|---|---|
| $20 \times 10$ | 100 | 100 |
| $20 \times 12$ | 94 | 95 |
| $20 \times 14$ | 100 | 100 |
| $20 \times 16$ | 73 | 73 |
| $20 \times 18$ | 99 | 100 |
| $20 \times 19$ | 100 | 100 |

**Table 4.6:** `Dualformer` (slow mode, trained on $20 \times 20$ mazes) generalizes to rectangular mazes.

### 4.1.5 ABLATION STUDY

As discussed in Section 3, the randomized traces for training `Dualformer` results from different trace dropping strategies, and there are numerous ways to combine them. We hereby ablate the design choices we made. First, to enable execution in both fast and slow modes, a naïve alternative approach is to train `Dualformer` using a mixture of solution-only and complete-trace data, i.e. $p_1 = p_2 = p_3 = 0$ in our randomization strategy. We refer to such variants as *Mix-p* models, where $p$ is the fraction of solution-only data in the training dataset. Note that the Solution-Only model is essentially a Mix-1 model, and the Complete-Trace model is a Mix-0 model. Below, we compare `Dualformer` to Mix-$p$ models for both inference modes. Second, our dropping strategies are

| | Method | Avg Trace Length | 1-Optimal-64 / 3-Optimal-64 | 1-Solved-64 / 3-Solved-64 | SWC | Diversity |
|---|---|---|---|---|---|---|
| Maze 15 x 15 | Dropping Level 1 | 396 | 99.5 / 98.4 | 99.9 / 99.4 | 0.998 | 11.94 |
| | Dropping Level 1 + 2 | 324 | 98.9 / 98.2 | 99.7 / 99.1 | 0.996 | 11.87 |
| | Dropping Level 1 + 2 + 3 | 287 | 99.8 / 99.3 | 100 / 99.9 | 0.999 | 12.71 |
| | Dualformer (slow) | 278 | 99.6 / 99.2 | 99.9 / 99.9 | 0.999 | 12.54 |
| Maze 20 x 20 | Dropping Level 1 | 563 | 99.1 / 96.7 | 99.9 / 97.9 | 0.996 | 15.79 |
| | Dropping Level 1 + 2 | 549 | 98.7 / 96.7 | 99.2 / 98.5 | 0.991 | 17.51 |
| | Dropping Level 1 + 2 + 3 | 542 | 99.5 / 99.9 | 100 / 99.7 | 1.00 | 19.51 |
| | Dualformer (slow) | 439 | 98.9 / 97.8 | 99.9 / 99.7 | 0.998 | 18.86 |
| Maze 25 x 25 | Dropping Level 1 | 735 | 97.9 / 93.4 | 99.0 / 97.1 | 0.989 | 19.46 |
| | Dropping Level 1 + 2 | 750 | 99.2 / 96.6 | 99.8 / 99.1 | 0.997 | 23.18 |
| | Dropping Level 1 + 2 + 3 | 594 | 98.2 / 95.6 | 99.5 / 98.7 | 0.994 | 23.1 |
| | Dualformer (slow) | 589 | 99.9 / 97.2 | 99.7 / 99.3 | 0.997 | 25.05 |
| Maze 30 x 30 | Dropping Level 1 | 1183 | 97.5 / 93.3 | 98.9 / 97.4 | 0.988 | 22.61 |
| | Dropping Level 1 + 2 | 966 | 98.1 / 93.4 | 99.6 / 97.7 | 0.995 | 25.47 |
| | Dropping Level 1 + 2 + 3 | 759 | 96.8 / 92.2 | 99.8 / 98.4 | 0.996 | 24.94 |
| | Dualformer (slow) | 854 | 97.6 / 93.2 | 99.5 / 98.2 | 0.993 | 25.77 |
| Sokoban | Dropping Level 1 | 3000 | 94.2 / 86.6 | 95.6 / 90.9 | 0.950 | 3.90 |
| | Dropping Level 1 + 2 | 2272 | 93.9 / 86.7 | 96.5 / 89.8 | 0.960 | 4.41 |
| | Dropping Level 1 + 2 + 3 | 1638 | 95.5 / 90.7 | 97.7 / 95.0 | 0.970 | 4.39 |
| | Dualformer (slow) | 1482 | 94.5 / 87.6 | 97.4 / 94.1 | 0.970 | 4.66 |

**Table 4.7:** Comparison of different combinations of trace randomization strategies.

structured in a hierarchical manner. For instance, Level 2 dropping is developed based on the Level 1 strategy. We investigate how the performance changes when we halt the process at a specific level.

**Comparison with Mix-$p$ Models** Figure F.1 compares the 1-Optimal-64 rate of `Dualformer` and Mix-$p$ models. We test 8 values of $p$: 0 (equivalent to the Complete-Trace model), $0.05, 0.1, 0.2, 0.4, 0.6, 0.8$, and $1.0$ (equivalent to the Solution-Only model). In both inference modes, `Dualformer` beats all the Mix-$p$ models for all the five tasks we consider. In fact, `Dualformer` also outperforms Mix-$p$ models in the other metrics we consider too, see Appendix F. In particular, it is the "fastest" one when operating in slow mode: it generates the shortest reasoning trace.

**Combination of Randomization Strategies** We compare `Dualformer` with its variants where we halt the dropping strategies at specific levels. For the Maze environments, we fix the probability of using the complete reasoning trace for a training example to be $p_0 = 0.5$ for all the variants and vary the other probabilities. For the Sokoban environments, we fix the probability of Level 1 dropping to be $p_0 = 0.05$. Table G.1 lists the probabilities of different dropping strategies we use for all the models, and Table 4.7 shows the results. It should be noted that the variants cannot operate in fast mode, since they are not trained on any Level 4 data. Therefore, we only report the slow mode performance. As we increase the strategy level, the length of the reasoning trace decreases. In regard to the other metrics, the performance of Level 1+2+3 model and `Dualformer` are comparable. However, `Dualformer` enjoys the advantage of shorter reasoning trace and the capability to function in fast mode.

## 4.2 APPLICATION TO LLM TRAINING: MATH REASONING

In this section, we show the efficacy of structured trace dropping techniques for training large-scale LLMs to solve math problems. Particularly, we finetune Llama-3-8B and Mistral-7B models using a dataset that contains a variety of math questions and answers with detailed reasoning steps, where we utilize a trace dropping technique that leverages the specific structure of the reasoning trace for math problems too. We benchmark the resulting models against corresponding base models finetuned on the dataset directly.

**Dataset** We evaluate all the models using a dataset named `Aug-MATH`, which is derived from the `MATH` dataset (Hendrycks et al., 2021) that contains 7500 training examples of math questions and solutions, and 5000 testing examples. Following Yu et al. (2023), we query the Llama-3.1-70B-Instruct model to rewrite the solutions to include more detailed intermediate steps, following a given format. To encourage the diversity of reasoning traces, we sample 4 LLM responses for each problem using a temperature of 0.7 and top-$p = 0.9$. The resulting dataset then contains 30000 training examples and 5000 testing examples. In Appendix H.1, we show the prompt template that we use for solution rewriting, and a concrete training example before and after rewriting.

**Structured Trace Dropping and Randomization** The answer of the math questions rewritten by Llama-3.1-70B-Instruct contains 6-10 intermediate reasoning steps on average, where every step might contain multiple sentences. We use a randomized training procedure similar to the framework proposed in Section 3. For each of the training examples sampled in a batch, we randomly drop each intermediate reasoning step with probability $p$. Below, we report the results when varying $p$.

**Hyperparameters** We finetune two base models Mistral-7B and LLama-3-8B for two epochs using a batch size of 32. We use the AdamW optimizer (Loshchilov & Hutter, 2019) with a learning rate of $8e{-}6$ for the Mistral models, and $5e{-}6$ for the Llama-3 models. Full training details, including how we select the learning rates and other hyperparameters, are deferred to Appendix C.2. For fine-tuning the models, we use the same prompt as in Yu et al. (2023), which is displayed in Appendix H.2.

**Evaluation** Similar to `Dualformer`, we evaluate the models in both the fast and the slow mode, where the LLM is required to output the final solution directly or to solve the problem step by step. Following Yu et al. (2023), we use zero-shot prompting when evaluating the models. See Appendix H.2 for the prompts we use in each mode. We consider the *Greedy@1* metric (Dubey et al., 2024; Yu et al., 2023), where for each question we generate 1 response using a temperature of 0 and verify the correctness. We also report the *pass@20* metric (Chen et al., 2021), where we randomly sample 20 responses using a temperature of 0.5. For reference, we also report the results of fine-tuning those models on the original `MATH` dataset.

**Results** The results are presented in Table 4.8. We test four values of $p$: $0.1, 0.2, 0.3$ and $0.4$. Our results show that the proposed training strategy makes both LLMs more effective and efficient. We first inspect the results of the Mistral-7B models. For the slow mode inference, fine-tuning the model with trace dropping and randomized training improves upon the baseline model that is directly finetuned on `Aug-MATH` dataset. The absolute Greedy@1 metric improved by 1.7% when $p = 0.1$ (which amounts to 10% relative performance improvement), and 0.9% with $p = 0.2$ and $0.3$, and 0.1% when $p = 0.4$. Our models are also outperforming the baseline model for the Pass@20 metric when $p = 0.1, 0.2$ and $0.3$, where the absolute correct rate increases to $61.9\%$. Under both evaluation schemes, the average length of the reasoning trace goes down as $p$ goes up. Similarly, for inference in the fast mode, our models also achieve higher correct rate. A similar trend of performance improvement also holds for the Llama-3-8B models. Finally, for reader's reference, we include the results of both the Mistral-7B and the Llama-3-8B models fine-tuned on the original `MATH` dataset, which clearly lags behind[2].

| Model | Dataset & Dropping Prob | Greedy@1(%) (slow / fast) | Trace Length | Pass@20(%) (slow / fast) | Trace Length |
|---|---|---|---|---|---|
| Mistral-7B | Aug-MATH (baseline) | 16.9 / 9.6 | 527 / - | 59.6 / 29.8 | 521 / - |
| | Aug-MATH (p=0.1) | 18.6 / 11.3 | 508 / - | 61.6 / 32.0 | 479 / - |
| | Aug-MATH (p=0.2) | 17.8 / 11.2 | 477 / - | 61.4 / 31.9 | 470 / - |
| | Aug-MATH (p=0.3) | 17.8 / 11.8 | 497 / - | 61.9 / 31.7 | 466 / - |
| | Aug-MATH (p=0.4) | 17.0 / 11.0 | 434 / - | 56.4 / 28.9 | 397 / - |
| | MATH | 13.1 / 8.5 | 290 / - | 53.0 / 29.4 | 227 / - |
| Llama-3-8B | Aug-MATH (baseline) | 19.7 / 13.1 | 548 / - | 62.7 / 35.6 | 535 / - |
| | Aug-MATH (p=0.1) | 20.1 / 13.3 | 544 / - | 63.4 / 36.2 | 522 / - |
| | Aug-MATH (p=0.2) | 20.5 / 13.8 | 525 / - | 63.9 / 36.7 | 497 / - |
| | Aug-MATH (p=0.3) | 20.5 / 13.5 | 515 / - | 63.4 / 37.5 | 474 / - |
| | Aug-MATH (p=0.4) | 20.4 / 13.5 | 490 / - | 63.4 / 37.2 | 450 / - |
| | MATH | 13.3 / 12.6 | 432 / - | 52.8 / 35.5 | 332 / - |

**Table 4.8:** The performance of Llama-3 and Mistral models finetuned to solve math problems.

## 5  CONCLUSION

We present a simple and easy-to-implement framework for training Transformers to solve reasoning and planning tasks. We carefully probe the structure of the reasoning traces and design corresponding dropping strategies that imitate the shortcuts in human thinking process. By randomly applying the dropping strategies to training examples, the resulting model, `Dualformer`, can be controlled to execute in either fast or slow reasoning mode, or in auto mode where it decides the mode to engage. `Dualformer` achieves enhanced performance for the maze navigation tasks and the Sokoban game, while reducing the number of reasoning steps required. Remarkably, our approach is not limited to training task-specific models from scratch. We apply techniques in the same spirit to fine-tune LLMs to answer math questions and obtain improved performance. Last, the proposed framework also reduces computation consumption as the input sequences are shortened after trace dropping. Future work could investigate whether our approach helps models scale, and explore methodologies such as curriculum learning and hierarchical planning to adapt `Dualformer` for more complex tasks.

---

[2]The trace generated by models trained on `MATH` is much shorter, because the answers in `MATH` do not contain as many intermediate steps as those in `Aug-MATH`.

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

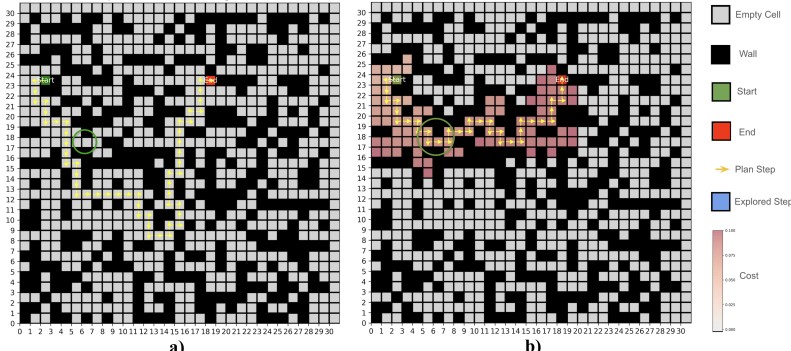

**Figure B.1:** The left panel plots a path generated by fast-thinking, which is suboptimal. The right panel plots a optimal path generated by slow-thinking. Cells highlighted in red are the positions visited by the reasoning trace of slow-thinking mode. Green circle shows a critical cell that needs to identified for a optimal path yet missed by the fast-thinking.

## A  PRELIMINARIES

Our work builds upon the work of Lehnert et al. (2024). To perform planning, we train a Transformer to model a sequence of tokens that sequentially represents the planning task, the computation of A* algorithm, and the optimal solution derived from the A* search. The tokenization method is illustrated in Figure A.1. As a toy example, we consider a navigation task in a $3 \times 3$ Maze where the goal is to find one shortest path from the start cell to goal cell, without hitting a wall cell. The A* algorithm has successfully determined an optimal plan. We use a sequence of tokens to express both the task and the Maze structure, which also serves as the *prompt* for Dualformer. The solution is depicted by the plan token sequence that describes the path using coordinates. The A* algorithm generates a search trace sequence that records the search dynamics performed, displayed in Figure 3.1. Recall that the A* algorithm is a pathfinding algorithm on a weighted graph. The create clause adds the node (represented by the subsequent coordinates) into the search frontier, and the close clause adds the node to the closed set. Each clause, either create or close, is followed by the tokens x, y, c0, and c1, representing the node's coordinates, cost-since-start value, and heuristic values, respectively. For details of A* and the tokenization approach, we refer the readers to Russell & Norvig (2021) and Lehnert et al. (2024), respectively.

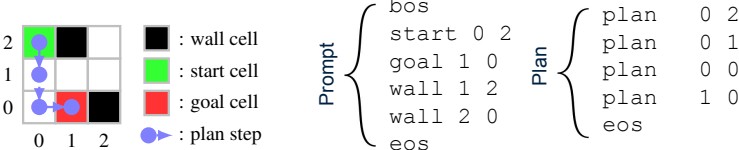

**Figure A.1:** An example Maze navigation task and one optimal plan obtained by the A* algorithm. The task and the plan are expressed as a prompt token sequence and a plan token sequence.

## B  COMPARISON OF FAST THINKING AND SLOW THINKING

## C  NETWORK ARCHITECTURE AND HYPERPARAMETERS

We use the same encoder-decoder Transformer architecture as in Lehnert et al. (2024) for Dualformer. It first converts each token into a one-hot vector, which is then transformed into a set of vectors through the embedding layer. The embedding vectors then go through the subsequent layers shown in Lehnert et al. (2024, Figure 4). We use RoPE embeddings for positional encoding, and no dropout is used in our architecture.

For the model size, architecture parameters, batch size, we use the same setup as in as (Lehnert et al., 2024). Specifically, for the Maze tasks, we use a 15M-parameter model which consists of 3 attention heads and 6 layers with hidden size 64. We optimize the model by the AdamW (Loshchilov & Hutter, 2019) optimizer with learning rate 2.5e-4 and batch size 16, where $\beta_0$ and $\beta_1$ set to 0.9 and 0.99,

respectively. A linear warm-up strategy was employed for the first 2000 gradient updates. Afterward, we use the cosine learning rate scheduler (Loshchilov & Hutter, 2016).

For the Sokoban tasks, we use a 46M-parameter model which consists of 4 attention heads and 8 layers with hidden size 96. We use batch size 64 and learning rate 7.5e-5, while the other hyperparameters are the same as above.

## C.1 ENVIRONMENT & DATASET

We used the same dataset as in Lehnert et al. (2024), which is available at `https://github.com/facebookresearch/searchformer`. The Maze dataset contains $10M$ examples and the Sokoban dataset contains $10M$ examples and we use the first $100k$ example sorted in accordance to their "id" field in mongodb (following same approach as (Lehnert et al., 2024)). For reader's reference, the dataset is generated as follows. For the Maze tasks, 30-50% of the cells were first randomly designated as walls. Next, a start and a goal location were chosen randomly. Then, the $A^*$ algorithm was applied to generate an optimal plan. For the Sokoban tasks, we use a $7 \times 7$ grid map where two wall cells were randomly inserted as obstacles. Moreover, two docks, two boxes, and two worker locations were placed randomly. Once a game is generated, it is only added to the dataset if it could be solved by the $A^*$ algorithm.

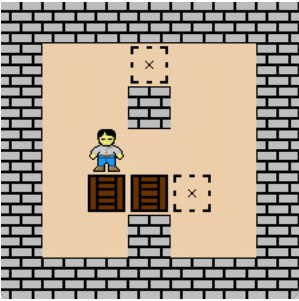

**Figure C.1:** Map of an example Sokoban task in a $7 \times 7$ grid world, with two boxes and two docks (figure taken from Lehnert et al. (2024)).The worker needs to push (and not pull) the boxes from the start locations to the destinations.

```
        ⎧ bos                          ⎧ bos                                              ⎧ plan 2 3
        │ worker 2 3                    │ create worker 2 3 box 2 4 box 3 4 c0  c3         │ plan 1 3
        │ box    2 4          Trace    ⎨ close  worker 2 3 box 2 4 box 3 4 c0  c3          │ plan 1 4
        │ box    3 4      (2583 tokens) │ ...                                              │ plan 1 5
        │ dock   1 3                    │ create worker 5 4 box 2 3          c10 c3        │ plan 2 5
Prompt ⎨ dock   4 4                     ⎩ close  worker 2 1                  c12 c0        │ plan 2 4
        │ wall   0 0                                                                  Plan⎨ plan 3 4
        │ wall   0 1                                                                      │ plan 2 4
        │ ...                                                                             │ plan 2 3
        │ wall   6 6                                                                      │ plan 2 2
        ⎩ eos                                                                             │ plan 1 2
                                                                                          │ plan 1 1
                                                                                          │ plan 2 1
                                                                                          ⎩ eos
```

**Figure C.2:** Example prompt and response token sequences for the Sokoban task depicted in Figure C.1 (example taken from (Lehnert et al., 2024)).

## C.2 MATH REASONING

We use the implementation provided at `https://github.com/meta-llama/llama-recipes` for fine-tuning the models.

We train all the models for 2 epochs, using a batch size of 32. We use the AdamW optimizer with a learning rate of $5 \times 10^{-6}$ for the Llama model and $8 \times 10^{-6}$ for the Mistral models. The learning rate is selected as follows. We sweep over 3 values $2 \times 10^{-6}, 5 \times 10^{-6}, 8 \times 10^{-6}$ and choose the

learning rate that yields the lowest validation loss. We then retrain the models using the selected learning rates on the full training dataset and report the results. We use the default values for the other hyperparameters. More specifically, we do not use linear rate warmup, weight decay, nor multistep gradient accumulation. We use betas=$(0.9, 0.999)$, eps=$1e - 8$, $\gamma = 0.85$ (multiplicative step-wise learning rate decay) for AdamW and "packing" as the batching strategy.

# D    CONTROLLABLE GENERATION OF DUALFORMER

`Dualformer` offers flexible output options for users to choose. In this section, we demonstrate this by an example navigation task in a $15 \times 15$ maze, see Figure D.1. The start location is (9, 10) and the goal location is (3, 6). The location coordinates and the maze structure is encoded in the original prompt below.

---

**Original Prompt**

```
bos, start, 9, 10, goal, 3, 6, wall, 0, 0, wall, 4, 0, wall,
7, 0, wall, 10, 0, wall, 12, 0, wall, 13, 0, wall, 3, 1,
wall, 7, 1, wall, 11, 1, wall, 12, 1, wall, 13, 1, wall, 14,
1, wall, 0, 2, wall, 3, 2, wall, 4, 2, wall, 6, 2, wall, 7,
2, wall, 8, 2, wall, 10, 2, wall, 11, 2, wall, 14, 2, wall,
1, 3, wall, 2, 3, wall, 3, 3, wall, 11, 3, wall, 13, 3, wall,
2, 4, wall, 8, 4, wall, 10, 4, wall, 11, 4, wall, 12, 4,
wall, 14, 4, wall, 3, 5, wall, 4, 5, wall, 5, 5, wall, 7, 5,
wall, 9, 5, wall, 11, 5, wall, 12, 5, wall, 14, 5, wall, 0,
6, wall, 4, 6, wall, 6, 6, wall, 8, 6, wall, 9, 6, wall, 14,
6, wall, 2, 7, wall, 4, 7, wall, 7, 7, wall, 9, 7, wall, 10,
7, wall, 13, 7, wall, 2, 8, wall, 3, 8, wall, 5, 8, wall, 6,
8, wall, 8, 8, wall, 10, 8, wall, 11, 8, wall, 12, 8, wall,
1, 9, wall, 5, 9, wall, 6, 9, wall, 9, 9, wall, 11, 9, wall,
14, 9, wall, 10, 10, wall, 12, 10, wall, 13, 10, wall, 14,
10, wall, 1, 11, wall, 8, 11, wall, 9, 11, wall, 12, 11,
wall, 3, 12, wall, 5, 12, wall, 6, 12, wall, 8, 12, wall,
10, 12, wall, 12, 12, wall, 14, 12, wall, 0, 13, wall, 1, 13,
wall, 3, 13, wall, 8, 13, wall, 12, 13, wall, 13, 13, wall,
14, 13, wall, 0, 14, wall, 1, 14, wall, 2, 14, wall, 6, 14,
wall, 14, 14, eos
```

---

To configure `Dualformer`'s operation mode, we only need to append `bos` and a control token to the original prompt. To output solution only (fast mode), we inject `plan`. On the other hand, we inject `create` to let `Dualformer` output both trace and solution.

## D.1    FAST MODE

The modified prompt for fast mode is displayed below. The extra tokens are highlighted in purple.

---

**Fast Mode Prompt (two extra tokens added to the original)**

```
bos, start, 9, 10, goal, 3, 6, wall, 0, 0, wall, 4, 0, wall,
7, 0, wall, 10, 0, wall, 12, 0, wall, 13, 0, wall, 3, 1,
wall, 7, 1, wall, 11, 1, wall, 12, 1, wall, 13, 1, wall, 14,
1, wall, 0, 2, wall, 3, 2, wall, 4, 2, wall, 6, 2, wall, 7,
2, wall, 8, 2, wall, 10, 2, wall, 11, 2, wall, 14, 2, wall,
1, 3, wall, 2, 3, wall, 3, 3, wall, 11, 3, wall, 13, 3, wall,
2, 4, wall, 8, 4, wall, 10, 4, wall, 11, 4, wall, 12, 4,
wall, 14, 4, wall, 3, 5, wall, 4, 5, wall, 5, 5, wall, 7, 5,
wall, 9, 5, wall, 11, 5, wall, 12, 5, wall, 14, 5, wall, 0,
6, wall, 4, 6, wall, 6, 6, wall, 8, 6, wall, 9, 6, wall, 14,
6, wall, 2, 7, wall, 4, 7, wall, 7, 7, wall, 9, 7, wall, 10,
7, wall, 13, 7, wall, 2, 8, wall, 3, 8, wall, 5, 8, wall, 6,
8, wall, 8, 8, wall, 10, 8, wall, 11, 8, wall, 12, 8, wall,
1, 9, wall, 5, 9, wall, 6, 9, wall, 9, 9, wall, 11, 9, wall,
14, 9, wall, 10, 10, wall, 12, 10, wall, 13, 10, wall, 14,
10, wall, 1, 11, wall, 8, 11, wall, 9, 11, wall, 12, 11,
wall, 3, 12, wall, 5, 12, wall, 6, 12, wall, 8, 12, wall,
10, 12, wall, 12, 12, wall, 14, 12, wall, 0, 13, wall, 1, 13,
wall, 3, 13, wall, 8, 13, wall, 12, 13, wall, 13, 13, wall,
14, 13, wall, 0, 14, wall, 1, 14, wall, 2, 14, wall, 6, 14,
wall, 14, 14, eos, bos, plan
```

The following box shows the generated tokens, and Figure D.1a plot the corresponding path.

---

**Fast Mode Output**

```
9, 10, plan, 8, 10, plan, 7, 10, plan, 6, 10, plan, 5, 10,
plan, 4, 10, plan, 3, 10, plan, 2, 10, plan, 1, 10, plan, 0,
10, plan, 0, 9, plan, 0, 8, plan, 0, 7, plan, 1, 7, plan, 1,
6, plan, 2, 6, plan, 3, 6, eos
```

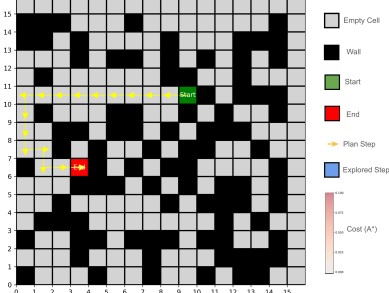

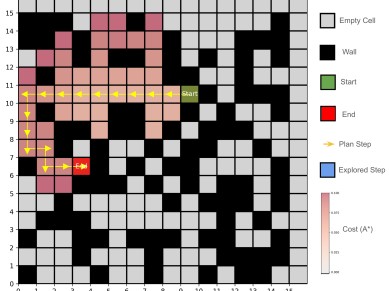

**(a)** `Dualformer` operates in fast-mode, directly outputting the final plan.

**(b)** `Dualformer` operates in slow-mode, generating search traces before the final solution.

**Figure D.1:** Controllable generation of `Dualformer`. Paths correspond to the output displayed in Appendix D. The wall cells are depicted in dark, while the unoccupied cells are shown in gray. The starting point is marked in green, and the destination point is in red. The final path determined by the trained model is highlighted in yellow. When `Dualformer` operates in slow mode, the corresponding cells explored by the search trace are highlighted in red, visually representing each step of the search process. The color intensity indicates the total cost value.

## D.2 SLOW MODE

Similarly, we present the modified prompt for slow mode below. Comparing with the fast mode prompt, the only change is the control token becomes `create`.

---
**Slow Mode Prompt (two extra tokens added to the original)**

```
bos, start, 9, 10, goal, 3, 6, wall, 0, 0, wall, 4, 0, wall,
7, 0, wall, 10, 0, wall, 12, 0, wall, 13, 0, wall, 3, 1,
wall, 7, 1, wall, 11, 1, wall, 12, 1, wall, 13, 1, wall, 14,
1, wall, 0, 2, wall, 3, 2, wall, 4, 2, wall, 6, 2, wall, 7,
2, wall, 8, 2, wall, 10, 2, wall, 11, 2, wall, 14, 2, wall,
1, 3, wall, 2, 3, wall, 3, 3, wall, 11, 3, wall, 13, 3, wall,
2, 4, wall, 8, 4, wall, 10, 4, wall, 11, 4, wall, 12, 4,
wall, 14, 4, wall, 3, 5, wall, 4, 5, wall, 5, 5, wall, 7, 5,
wall, 9, 5, wall, 11, 5, wall, 12, 5, wall, 14, 5, wall, 0,
6, wall, 4, 6, wall, 6, 6, wall, 8, 6, wall, 9, 6, wall, 14,
6, wall, 2, 7, wall, 4, 7, wall, 7, 7, wall, 9, 7, wall, 10,
7, wall, 13, 7, wall, 2, 8, wall, 3, 8, wall, 5, 8, wall, 6,
8, wall, 8, 8, wall, 10, 8, wall, 11, 8, wall, 12, 8, wall,
1, 9, wall, 5, 9, wall, 6, 9, wall, 9, 9, wall, 11, 9, wall,
14, 9, wall, 10, 10, wall, 12, 10, wall, 13, 10, wall, 14,
10, wall, 1, 11, wall, 8, 11, wall, 9, 11, wall, 12, 11,
wall, 3, 12, wall, 5, 12, wall, 6, 12, wall, 8, 12, wall,
10, 12, wall, 12, 12, wall, 14, 12, wall, 0, 13, wall, 1, 13,
wall, 3, 13, wall, 8, 13, wall, 12, 13, wall, 13, 13, wall,
14, 13, wall, 0, 14, wall, 1, 14, wall, 2, 14, wall, 6, 14,
wall, 14, 14, eos, bos, create
```
---

The following box shows the generated output, and Figure D.1b plots the corresponding trace and final path.

---
**Slow Mode Output**

```
9, 10, c0, c10, create, 8, 10, c1, c9, create, 8, 9, c2, c8,
create, 7, 10, c2, c8, create, 7, 9, c3, c7, create, 7, 8,
c4, c6, create, 6, 10, c3, c7, create, 7, 11, c3, c9, create,
5, 10, c4, c6, create, 6, 11, c4, c8, create, 5, 11, c5,
c7, create, 4, 10, c5, c5, create, 4, 11, c6, c6, create,
3, 10, c6, c4, create, 4, 9, c6, c4, create, 3, 9, c7, c3,
create, 4, 8, c7, c3, create, 2, 10, c7, c5, create, 3, 11,
c7, c5, create, 2, 9, c8, c4, create, 2, 11, c8, c6, create,
1, 10, c8, c6, create, 4, 12, c7, c7, create, 7, 12, c4, c10,
create, 7, 13, c5, c11, create, 0, 10, c9, c7, create, 4,
13, c8, c8, create, 2, 12, c9, c7, create, 7, 14, c6, c12,
create, 6, 13, c6, c10, create, 0, 9, c10, c6, create, 0,
11, c10, c8, create, 0, 8, c11, c5, create, 1, 12, c10, c8,
create, 2, 13, c10, c8, create, 4, 14, c9, c9, create, 5, 13,
c9, c9, create, 0, 7, c12, c4, create, 1, 8, c12, c4, create,
1, 7, c13, c3, create, 5, 13, c7, c9, create, 1, 6, c14, c2,
create, 5, 14, c8, c10, create, 1, 5, c15, c3, create, 2, 6,
c15, c1, create, 2, 5, c16, c2, create, 3, 6, c16, c0, plan,
9, 10, plan, 8, 10, plan, 7, 10, plan, 6, 10, plan, 5, 10,
plan, 4, 10, plan, 3, 10, plan, 2, 10, plan, 1, 10, plan, 0,
10, plan, 0, 9, plan, 0, 8, plan, 0, 7, plan, 1, 7, plan, 1,
6, plan, 2, 6, plan, 3, 6, eos
```
---

# E  DIVERSITY OF GENERATED PLANS

The `Dualformer` outperforms baseline models in discovering unique feasible solutions. To illustrate this visually, we select one example maze task and generate 64 responses using `Dualformer` in fast mode. Figure E.1 plots all the unique feasible paths discovered by fast mode `Dualformer` alongside those found by the Solution-Only baseline (64 responses). `Dualformer` (fast mode) identified 42 unique feasible paths, while the Solution-Only model only found 3. Similarly, Figure E.2 compares slow mode `Dualformer` and the Complete-Trace (`Searchformer`) baseline. `Dualformer` (slow mode) discovered 39 unique feasible paths, whereas the Complete-Trace model only found 17.

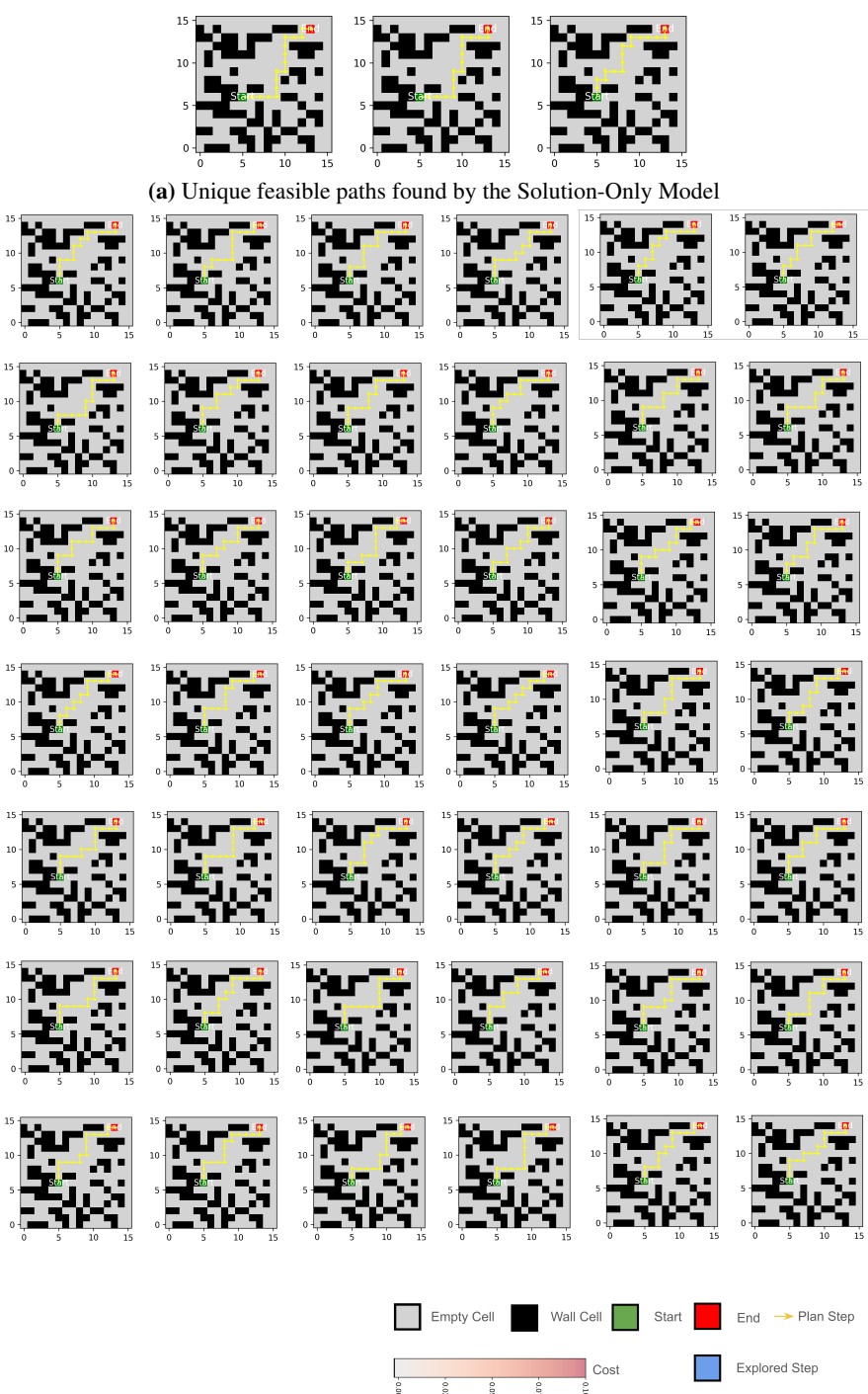

**(a)** Unique feasible paths found by the Solution-Only Model

**(b)** Unique feasible paths found by `Dualformer` (fast mode)

**Figure E.1:** Fast mode `Dualformer` finds more feasible paths than the Solution-Only model.The wall cells are depicted in dark, while the unoccupied cells are shown in gray. The starting point is marked in green, and the destination point is in red. The final path determined by the trained model is highlighted in yellow.

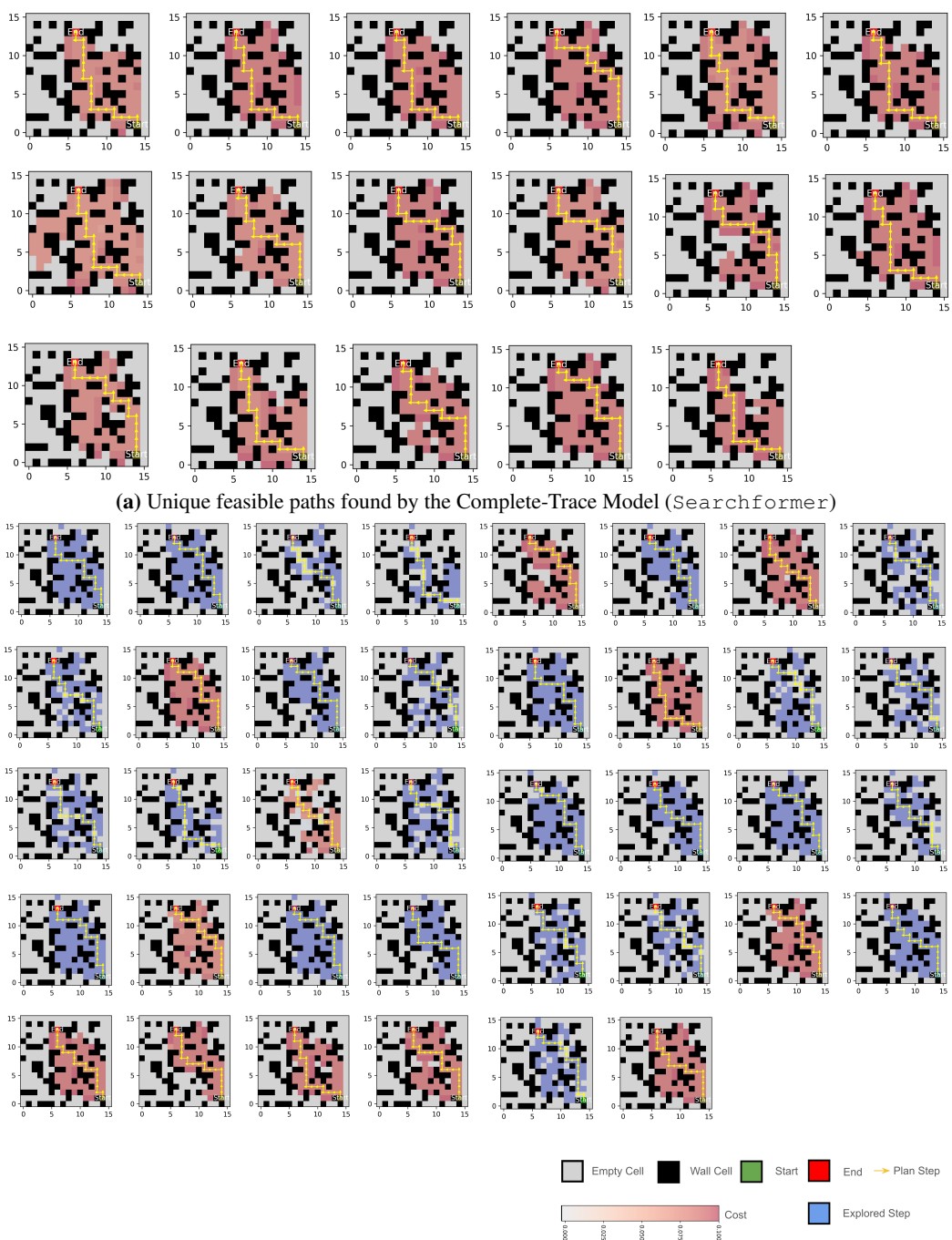

**(a)** Unique feasible paths found by the Complete-Trace Model (`Searchformer`)

**(b)** Unique feasible paths found by `Dualformer` (slow mode)

**Figure E.2:** Slow mode `Dualformer` finds more feasible paths than the Complete-Trace model (`Searchformer`). The wall cells are depicted in dark, while the unoccupied cells are shown in gray. The starting point is marked in green, and the destination point is in red. The final path determined by the trained model is highlighted in yellow. The cells explored by the search trace are highlighted in red, where the color intensity indicates the total cost value. If the generated search trace does not contain cost values (used in Level 2, 3, 4 dropping, Section 3), the explored cells are indicated in blue.

# F    COMPARISON WITH MIX-$p$ MODELS

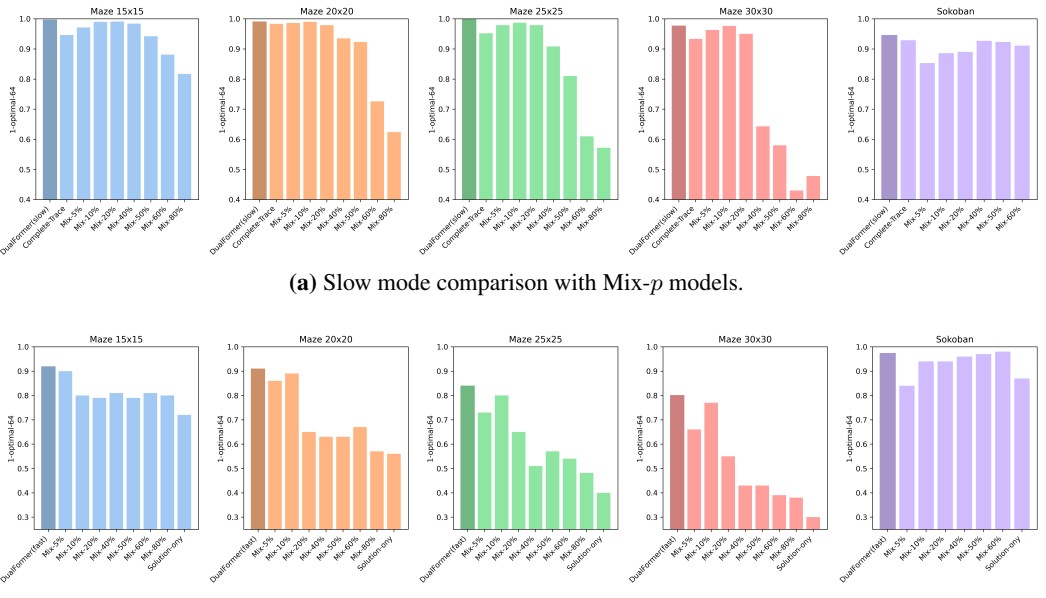

**(a)** Slow mode comparison with Mix-$p$ models.

**(b)** Fast mode comparison with Mix-$p$ models.

**Figure F.1:** The 1-Optimal-64 rate of `Dualformer` and Mix-$p$ models with varying values of $p$, where $p$ is the fraction of solution-only data in the corresponding training dataset. The top and bottom panels plot the results in fast and slow mode, respectively. `Dualformer` outperforms all the Mix-$p$ models in both inference modes. The probabilities of different dropping strategies used for `Dualformer` is described in the Hyperparameter paragraph, see Section 4.1.

| | Method | Avg Trace Length | 1-Optimal-64/ 5-Optimal-64 | 1-Solved-64 / 5-Solved-64 | SWC | Diversity |
|---|---|---|---|---|---|---|
| Maze 15 x 15 | Complete-Trace | 495 | 94.6 / 90.1 | 96.7 /93.0 | 0.964 | 7.60 |
| | Dualformer (slow) | 278 | 99.6 / 99.2 | 99.9 / 99.9 | 0.999 | 12.54 |
| | Mix-5% | 506 | 97.1 / 94.8 | 97.6 / 96.6 | 0.98 | 10.25 |
| | Mix-10% | 542 | 99.0 / 97.4 | 99.4 / 98.7 | 0.99 | 11.51 |
| | Mix-20% | 525 | 99.1 / 97.7 | 99.7 / 99.1 | 1.00 | 12.10 |
| | Mix-40% | 556 | 98.4 / 96.9 | 99.5 / 98.8 | 0.99 | 11.77 |
| | Mix-50% | 527 | 94.2 / 92.1 | 96.7 / 95.0 | 0.96 | 10.75 |
| | Mix-60% | 577 | 88.1 / 85.6 | 94.2 / 92.1 | 0.93 | 6.48 |
| | Mix-80% | 540 | 81.7 / 78.8 | 89.9 / 88.1 | 0.88 | 2.37 |
| Maze 20 x 20 | Complete-Trace | 851 | 98.3 / 95.5 | 98.8 / 93.0 | 0.987 | 14.53 |
| | Dualformer (slow) | 439 | 98.9 / 97.8 | 99.9 / 99.7 | 0.998 | 18.86 |
| | Mix-5% | 789 | 98.6 / 98.1 | 99.5 / 99.1 | 0.99 | 17.71 |
| | Mix-10% | 866 | 98.9 / 98.2 | 99.8 / 99.3 | 1.00 | 18.52 |
| | Mix-20% | 838 | 97.9 / 95.9 | 99.4 / 98.6 | 0.99 | 17.57 |
| | Mix-40% | 900 | 93.5 / 90.4 | 97.4 / 95.1 | 0.97 | 16.51 |
| | Mix-50% | 905 | 92.3 / 89.1 | 96.3 / 94.3 | 0.96 | 16.30 |
| | Mix-60% | 891 | 72.6 / 67.9 | 86.5 / 82.7 | 0.84 | 3.92 |
| | Mix-80% | 943 | 62.4 / 56.4 | 80.9 / 75.4 | 0.77 | 2.62 |
| Maze 25 x 25 | Complete-Trace | 1208 | 95.2 / 85.7 | 97.0 / 90.4 | 0.968 | 18.85 |
| | Dualformer (slow) | 589 | 99.9 / 97.2 | 99.7 / 99.3 | 0.997 | 25.05 |
| | Mix-5% | 1109 | 97.9 / 95.3 | 99.0 / 97.2 | 0.99 | 21.00 |
| | Mix-10% | 1264 | 98.7 / 96.3 | 99.5 / 98.0 | 0.99 | 22.33 |
| | Mix-20% | 1283 | 97.9 / 95.0 | 99.1 / 97.9 | 0.99 | 23.68 |
| | Mix-40% | 1278 | 90.8 / 87.4 | 96.3 / 93.9 | 0.96 | 22.30 |
| | Mix-50% | 1334 | 81.0 / 75.5 | 93.3 / 88.9 | 0.91 | 18.02 |
| | Mix-60% | 1266 | 61.0 / 53.4 | 80.3 / 74.0 | 0.77 | 4.63 |
| | Mix-80% | 1501 | 57.2 / 50.3 | 78.5 / 70.3 | 0.75 | 3.76 |
| Maze 30 x 30 | Complete-Trace | 1538 | 93.3 / 82.4 | 95.9 / 88.1 | 0.964 | 7.60 |
| | Dualformer (slow) | 854 | 97.6 / 93.2 | 99.5 / 98.2 | 0.993 | 25.77 |
| | Mix-5% | 2022 | 96.3 / 92.0 | 98.4 / 96.0 | 0.98 | 24.50 |
| | Mix-10% | 1720 | 97.7 / 95.0 | 99.2 / 98.2 | 0.99 | 27.17 |
| | Mix-20% | 1851 | 95.0 / 91.4 | 98.3 / 96.2 | 0.98 | 25.02 |
| | Mix-40% | 1854 | 64.3 / 56.2 | 83.1 / 76.0 | 0.81 | 12.60 |
| | Mix-50% | 1652 | 58.0 / 50.8 | 78.6 / 71.9 | 0.76 | 9.66 |
| | Mix-60% | 1983 | 43.0 / 35.4 | 68.4 / 58.1 | 0.65 | 3.21 |
| | Mix-80% | 1648 | 47.8 / 38.4 | 71.9 / 61.9 | 0.68 | 3.98 |
| Sokoban | Complete-Trace | 3600 | 92.9 / 84.4 | 94.7 / 89.0 | 0.944 | 2.91 |
| | Dualformer (slow) | 1482 | 94.5 / 87.6 | 97.4 / 94.1 | 0.97 | 4.66 |
| | Mix-5% | 3278 | 85.3 / 72.7 | 91.0 / 80.9 | 0.90 | 3.18 |
| | Mix-10% | 3402 | 88.6 / 77.2 | 94.1 / 87.9 | 0.93 | 4.07 |
| | Mix-20% | 3331 | 89.0 / 81.3 | 95.7 / 89.1 | 0.95 | 4.22 |
| | Mix-40% | 3294 | 92.7 / 86.1 | 97.1 / 93.1 | 0.96 | 4.14 |
| | Mix-50% | 3202 | 92.3 / 87.3 | 96.2 / 93.4 | 0.96 | 4.66 |
| | Mix-60% | 2594 | 91.1 / 83.2 | 96.4 / 91.0 | 0.96 | 4.48 |

**Table F.1:** Comparison of `Dualformer` and Mix-$p$ models in the slow mode.

| | Method | 1-Optimal-64 / 3-Optimal-64 | 1-Solved-64 / 3-Solved-64 | SWC | Diversity |
|---|---|---|---|---|---|
| | Dualformer (fast) | 91.8 / 87.6 | 97.1 / 94.8 | 0.960 | 9.05 |
| | Mix-5% | 89.8 / 83.2 | 92.7 / 89.0 | 0.92 | 11.72 |
| | Mix-10% | 80.3 / 75.6 | 90.5 / 86.9 | 0.88 | 7.01 |
| Maze 15x15 | Mix-20% | 78.8 / 77.6 | 87.8 / 86.4 | 0.86 | 1.97 |
| | Mix-40% | 81.2 / 78.8 | 88.6 / 86.9 | 0.87 | 1.73 |
| | Mix-50% | 79.0 / 76.4 | 87.7 / 85.8 | 0.86 | 1.92 |
| | Mix-60% | 81.3 / 78.0 | 89.3 / 87.3 | 0.88 | 1.95 |
| | Mix-80% | 79.8 / 76.6 | 88.6 / 86.0 | 0.87 | 2.12 |
| | Solution-Only | 72.0 / 68.9 | 82.7 / 80.1 | 0.80 | 1.52 |
| | Dualformer (fast) | 90.9 / 84.0 | 97.0 / 94.0 | 0.960 | 17.27 |
| | Mix-5% | 86.2 / 74.8 | 94.0 / 87.7 | 0.93 | 15.92 |
| | Mix-10% | 88.6 / 81.4 | 95.2 / 91.7 | 0.94 | 16.04 |
| Maze 20x20 | Mix-20% | 65.2 / 60.5 | 81.7 / 77.1 | 0.79 | 2.72 |
| | Mix-40% | 63.2 / 59.0 | 80.4 / 77.2 | 0.78 | 2.65 |
| | Mix-50% | 63.3 / 57.9 | 79.6 / 76.0 | 0.77 | 2.39 |
| | Mix-60% | 66.8 / 63.6 | 82.4 / 80.1 | 0.80 | 2.50 |
| | Mix-80% | 56.5 / 51.8 | 74.5 / 70.2 | 0.71 | 2.13 |
| | Solution-Only | 56.3 / 52.0 | 71.9 / 67.5 | 0.69 | 1.52 |
| | Dualformer (fast) | 83.9 / 72.9 | 95.5 / 90.6 | 0.940 | 21.23 |
| | Mix-5% | 73.4 / 56.9 | 87.0 / 77.3 | 0.85 | 13.37 |
| | Mix-10% | 80.2 / 68.0 | 90.4 / 83.1 | 0.89 | 16.19 |
| Maze 25x25 | Mix-20% | 64.8 / 57.5 | 85.0 / 78.0 | 0.81 | 10.51 |
| | Mix-40% | 50.7 / 46.3 | 73.8 / 68.9 | 0.69 | 3.60 |
| | Mix-50% | 56.5 / 50.8 | 77.9 / 71.9 | 0.74 | 3.15 |
| | Mix-60% | 53.6 / 48.9 | 75.9 / 69.8 | 0.72 | 3.07 |
| | Mix-80% | 48.2 / 44.2 | 69.6 / 64.4 | 0.66 | 2.88 |
| | Solution-Only | 39.7 / 34.7 | 60.3 /55.4 | 0.57 | 1.9 |
| | Dualformer (fast) | 80.0 / 66.0 | 91.8 / 85.7 | 0.906 | 18.23 |
| | Mix-5% | 66.3 / 46.1 | 83.6 / 72.0 | 0.82 | 11.85 |
| | Mix-10% | 76.9 / 65.8 | 90.8 / 84.7 | 0.89 | 19.35 |
| Maze 30x30 | Mix-20% | 55.4 / 43.5 | 78.3 / 67.4 | 0.75 | 9.97 |
| | Mix-40% | 43.4 / 37.3 | 69.7 / 63.2 | 0.66 | 3.79 |
| | Mix-50% | 43.0 / 38.7 | 68.1 / 62.9 | 0.64 | 3.23 |
| | Mix-60% | 39.2 / 33.8 | 65.2 / 59.0 | 0.61 | 3.09 |
| | Mix-80% | 38.0 / 32.3 | 62.2 / 55.5 | 0.58 | 2.75 |
| | Solution-Only | 30.0 / 26.0 | 54.1 / 47.8 | 0.50 | 1.86 |
| | Dualformer (fast) | 97.3 / 94.4 | 94.8 / 90.0 | 0.970 | 4.92 |
| | Mix-5% | 84.0 / 73.0 | 91.0 / 84.0 | 0.90 | 4.30 |
| | Mix-10% | 94.0 / 87.0 | 97.0 / 94.0 | 0.97 | 5.58 |
| Sokoban | Mix-20% | 94.0 / 89.0 | 97.0 / 95.0 | 0.97 | 4.10 |
| | Mix-40% | 96.0 / 93.0 | 98.0 / 97.0 | 0.98 | 4.15 |
| | Mix-50% | 97.0 / 96.0 | 99.0 / 98.0 | 0.99 | 3.38 |
| | Mix-60% | 98.0 / 97.0 | 99.0 / 99.0 | 0.99 | 3.25 |
| | solution-only | 86.8 / 83.4 | 92.8 / 90.0 | 0.92 | 1.24 |

**Table F.2:** Comparison of `Dualformer` and Mix-$p$ models in the fast mode.

# G  COMPARISON WITH DIFFERENT RANDOMIZATION STRATEGIES

In Table G.1 below, we show the probabilities of different dropping strategies we use for comparing different randomization strategies.

| | Model | Probabilities of Dropping Strategies |
|---|---|---|
| Maze | Dropping Level 1 | $p_0 = 0.5, p_1 = 0.5, p_2 = p_3 = p_4 = 0$ |
| | Dropping Level 1 + 2 | $p_0 = 0.5, p_1 = p_2 = 0.25, p_3 = p_4 = 0$ |
| | Dropping Level 1 + 2 + 3 | $p_0 = 0.5, p_1 = p_2 = p_3 = 1/6, p_4 = 0$ |
| | Dualformer | $p_0 = 0.45, p_1 = p_2 = p_3 = 1/6, p_4 = 0.05$ |
| Sokoban | Dropping Level 1 | $p_0 = 0.95, p_1 = 0.05, p_2 = p_3 = p_4 = 0$ |
| | Dropping Level 1 + 2 | $p_0 = 0.85, p_1 = 0.05, p_2 = 0.1, p_3 = p_4 = 0$ |
| | Dropping Level 1 + 2 + 3 | $p_0 = 0.75, p_1 = 0.05, p_2 = p_3 = 0.1, p_4 = 0$ |
| | Dualformer | $p_0 = 0.7, p_1 = 0.05, p_2 = p_3 = 0.1, p_4 = 0.05$ |

**Table G.1:** The probabilities of the dropping strategies used for each model.

## H DETAILS OF THE MATH REASONING EXPERIMENTS

### H.1 SOLUTION REWRITING

#### H.1.1 SOLUTION REWRITING PROMPT

We use the following prompt template for rewriting the solutions via LLama-3.1-70B-Instruct.

---

**Solution Rewriting Prompt**

I have a math problem and an initial chain of thought reasoning that needs elaboration. Please provide a more detailed step-by-step explanation for each part of the reasoning, including intermediate steps, calculations, and rationales behind each decision. Also, suggest any additional insights that might be relevant. Problem: {Question}? Current Chain of Thought: {CoT} The answer is {Final-answer}. Keep the total response less than 2048 tokens. Please expand on the above reasoning and give detailed reasoning chain and explanation. Write your answer in this format: "Here is a detailed step-by-step explanation of the reasoning: Reasons: <reason step >. eg.Step 1..Step 2...Step 3...Step N... The answer is: <answer >." Remember, always conclude with "\n\nThe answer is: <answer >."

---

#### H.1.2 SOLUTION REWRITING EXAMPLES

Below, we provide an example of a math question along with four generated solutions that have been rewritten by the Llama-3.1-70B-Instruct model. It is evident from this example that the original math dataset lacks sufficiently detailed solution steps required to solve the question, such as the formula for a geometric series.

---

**Original Training Example from** `MATH`

---

**Question**

The first term of a geometric sequence is 729, and the 7th term is 64. What is the positive, real value for the 5th term?

---

**Answer**

The only positive, real common ratio for this sequence is $\frac{2}{3}$. Thus, if $x$ is the 5th term, then $\left(\frac{2}{3}\right)^2 x = 64$, so $x = \boxed{144}$.

---

---

**Rewritten Answers in `Aug-MATH`**

---

**Llama-3.1-70B-Instruct Rewriting 1**

Here is a detailed step-by-step explanation of the reasoning:

\n\nStep 1: Recall the formula for the nth term of a geometric sequence. The nth term of a geometric sequence is given by the formula: $a_n = ar^{(n-1)}$, where $a$ is the first term, $r$ is the common ratio, and $n$ is the term number.

\n\nStep 2: Use the given information to set up an equation. We are given that the first term ($a$) is 729 and the 7th term ($a_7$) is 64. We can use this information to set up an equation using the formula from Step 1: $64 = 729r^{(7-1)}$.

\n\nStep 3: Simplify the equation. Simplifying the equation from Step 2, we get: $64 = 729r^6$.

\n\nStep 4: Solve for the common ratio ($r$). To solve for $r$, we can divide both sides of the equation by 729 and take the 6th root of both sides: $r = \left(\frac{64}{729}\right)^{\frac{1}{6}}$.

\n\nStep 5: Calculate the value of $r$. Evaluating the expression from Step 4, we get: $r = \left(\frac{64}{729}\right)^{\frac{1}{6}} = \left(\frac{2}{3}\right)^2 = \frac{2}{3}$.

\n\nStep 6: Use the value of $r$ to find the 5th term. Now that we have the value of $r$, we can use the formula from Step 1 to find the 5th term ($a_5$): $a_5 = 729\left(\frac{2}{3}\right)^{(5-1)}$.

\n\nStep 7: Simplify the expression for $a_5$. Simplifying the expression from Step 6, we get: $a_5 = 729\left(\frac{2}{3}\right)^4 = 729\left(\frac{16}{81}\right) = 144$.

\n\nThe answer is: 144.

---

**Llama-3.1-70B-Instruct Rewriting 2**

Here is a detailed step-by-step explanation of the reasoning:

\n\nStep 1: Understand the problem The problem presents a geometric sequence with a given first term (729) and a given 7th term (64). We need to find the positive, real value of the 5th term.

\n\nStep 2: Identify the common ratio Since it's a geometric sequence, we know that each term is obtained by multiplying the previous term by a fixed constant, called the common ratio (r). We need to find this common ratio.

\n\nStep 3: Use the given terms to set up an equation We know that the 7th term is 64, and the first term is 729. Using the formula for the nth term of a geometric sequence: $a_n = a_1 \cdot r^{n-1}$, we can set up the equation: $64 = 729 \cdot r^6$.

\n\nStep 4: Solve for the common ratio Divide both sides by 729: $\frac{64}{729} = r^6$. Take the 6th root of both sides: $r = \sqrt[6]{\frac{64}{729}} = \frac{2}{3}$.

\n\nStep 5: Verify the positive, real common ratio We are told that the only positive, real common ratio for this sequence is $\frac{2}{3}$. This is indeed true, as the 6th root of $\frac{64}{729}$ is $\frac{2}{3}$.

\n\nStep 6: Use the common ratio to find the 5th term Now that we have the common ratio, we can find the 5th term. Let $x$ be the 5th term. Using the formula for the nth term: $x = 729 \cdot \left(\frac{2}{3}\right)^4$.

\n\nStep 7: Simplify the expression $x = 729 \cdot \frac{16}{81} = 144$.

\n\nStep 8: Verify the answer We can verify that $\left(\frac{2}{3}\right)^2 x = 64$ indeed holds true for $x = 144$. \n\nThe answer is: $\boxed{144}$

Rewritten Answers in `Aug-MATH`

### Llama-3.1-70B Instruct Rewriting 3

Here is a detailed step-by-step explanation of the reasoning: \n\nStep 1: Understand the problem We are given a geometric sequence with a first term of 729 and a 7th term of 64. We need to find the positive, real value of the 5th term.

\n\nStep 2: Recall the formula for a geometric sequence A geometric sequence is defined by the formula: $a_n = a_1 \cdot r^{n-1}$, where $a_n$ is the nth term, $a_1$ is the first term, $r$ is the common ratio, and $n$ is the term number. \n\nStep 3: Use the given information to set up an equation We know the first term ($a_1 = 729$) and the 7th term ($a_7 = 64$). We can use this information to set up an equation: $64 = 729 \cdot r^{7-1}$, which simplifies to $64 = 729 \cdot r^6$.

\n\nStep 4: Solve for the common ratio (r) To find the common ratio, we can divide both sides of the equation by 729, resulting in $\frac{64}{729} = r^6$. Taking the 6th root of both sides, we get $r = \left(\frac{64}{729}\right)^{\frac{1}{6}}$.

\n\nStep 5: Simplify the common ratio Evaluating the expression, we find that $r = \frac{2}{3}$.

\n\nStep 6: Use the common ratio to find the 5th term Now that we have the common ratio, we can use it to find the 5th term. We know that $a_5 = a_1 \cdot r^{5-1} = 729 \cdot \left(\frac{2}{3}\right)^4$.

\n\nStep 7: Simplify the expression for the 5th term Evaluating the expression, we get $a_5 = 729 \cdot \frac{16}{81} = 144$.

\n\nThe answer is: 144.

### Llama-3.1-70B-Instruct Rewriting 4

Here is a detailed step-by-step explanation of the reasoning:
Reasons:

\n\nStep 1: The problem provides us with the first term of a geometric sequence, which is 729, and the 7th term, which is 64. We are asked to find the positive, real value for the 5th term.

\n\nStep 2: To solve this problem, we need to find the common ratio of the geometric sequence. We know that the common ratio is the ratio of any term to its previous term. Let's denote the common ratio as $r$.

\n\nStep 3: Using the given information, we can set up an equation to find the common ratio. Since the 7th term is 64, we can write: $729 \cdot r^6 = 64$.

\n\nStep 4: Now, we can solve for $r$ by taking the 6th root of both sides of the equation: $r = \left(\frac{64}{729}\right)^{\frac{1}{6}}$.

\n\nStep 5: Simplifying the expression, we get: $r = \left(\frac{2}{3}\right)^2 = \frac{4}{9}$. However, this is not the only possible value for the common ratio. We need to check if there are other positive, real values of $r$ that satisfy the equation.

\n\nStep 6: We can do this by factoring 729 and 64 to see if there are any common factors that could lead to other possible values of $r$. Factoring 729, we get: $729 = 3^6$. Factoring 64, we get: $64 = 2^6$.

\n\nStep 7: Since 729 and 64 have no common factors other than 1, we can conclude that the only positive, real common ratio for this sequence is indeed $\left(\frac{2}{3}\right)^2 = \frac{4}{9}$.

\n\nStep 8: Now that we have found the common ratio, we can use it to find the 5th term of the sequence. Let's denote the 5th term as $x$. We know that the 5th term is obtained by multiplying the 3rd term by the common ratio twice. Since the 3rd term is $729 \cdot \left(\frac{2}{3}\right)^2 = 144$, we can write: $x = 144 \cdot \left(\frac{2}{3}\right)^2$.

\n\nStep 9: Simplifying the expression, we get: $x = 144$.

\n\nThe answer is: 144.

## H.2 FINE-TUNING AND EVALUATION PROMPTS

We use the following prompt for fine-tuning both the Mistral and the Llama model, following (Yu et al., 2023).

---

**Fintuning Prompt**

<start-header-token>Below is an instruction that describes a task. Write a response that appropriately completes the request. \n\n ### Instruction:\n{Question} \n\n ### Response: Let's think step by step. {CoT steps + final solution}

---

We use the following prompt for the slow mode evalaution.

---

**Slow Mode Evaluation Prompt**

<start-header-token>Below is an instruction that describes a task. Write a response that appropriately completes the request. \n\n ### Instruction:\n{Question} \n\n ### Response: Let's think step by step.

---

For fast mode evaluation, we force the generation to directly output the final answer by adding the phrase "The answer is: ".

---

**Fast Mode Evaluation Prompt**

<start-header-token>Below is an instruction that describes a task. Write a response that appropriately completes the request. \n\n ### Instruction:\n{Question} \n\n ### Response: \n\nThe answer is:

---

## H.3 EXAMPLE OUTPUTS

We provide a few example math questions and answers randomly sampled from our trained model, alongside the answers output by the baseline model. Within each answer box, on the left is the baseline model output (long and lengthy) and on the right is our method output (effective and efficient, colored in blue).

---

**Question 1**

In Pascal's Triangle, the third row has four elements, and the sum of these elements is 8. What is the sum of the elements in the tenth row? (Note that we are counting the top row with a single 1 as the 0th row).

---

---

**Answers to Q1**

**Llama-3-Aug (baseline)** Here is a detailed step-by-step explanation of the reasoning:

- Step 1: Understand the problem and the given information. We are asked to find the sum of the elements in the tenth row of Pascal's Triangle, given that the third row has four elements and their sum is 8.

- Step 2: Recall the formula for the sum of the elements in a row of Pascal's Triangle. The sum of the elements in the nth row is given by the formula: $\binom{n}{0} + \binom{n}{1} + \binom{n}{2} + \cdots + \binom{n}{n} = 2^n$.

- Step 3: Identify the relationship between the sum of the elements in a row and the sum of the elements in the previous row. Since the sum of the elements in the nth row is $2^n$, the sum of the elements in the (n-1)th row is $2^{n-1}$.

- Step 4: Recognize that the sum of the elements in the third row is 8, which is equal to $2^3$. This is because the sum of the elements in the third row is $\binom{3}{0} + \binom{3}{1} + \binom{3}{2} = 1 + 3 + 3 = 7$, and the extra 1 comes from the top row, which is counted as the 0th row.

- Step 5: Use the relationship between the sums of the elements in consecutive rows to find the sum of the elements in the tenth row. Since the sum of the elements in the third row is $2^3$, and the sum of the elements in the (n-1)th row is $2^{n-1}$, we can set up the equation: $2^3 = 2^{10-1}$.

- Step 6: Solve the equation for n. Taking the logarithm of both sides, we get: $3 \log 2 = (10 - 1) \log 2$. Simplifying, we get: $3 \log 2 = 9 \log 2$, which implies that $\log 2 = 0$. This is a contradiction, since $\log 2$ is a positive number.

- Step 7: Conclude that the equation $2^3 = 2^{10-1}$ has no solution, and therefore, the sum of the elements in the tenth row is not equal to $2^{10-1}$.

- Step 8: Recognize that the sum of the elements in the tenth row is still given by the formula: $\binom{10}{0} + \binom{10}{1} + \binom{10}{2} + \cdots + \binom{10}{10} = 2^{10}$.

The answer is: 1024.

**Llama-3-Aug (p=0.4, ours)** Here is a detailed step-by-step explanation of the reasoning:

- Step 1: Understand the problem and the given information. We are asked to find the sum of the elements in the tenth row of Pascal's Triangle, given that the third row has four elements and their sum is 8.

- Step 2: Recall the properties of Pascal's Triangle. The elements in each row of Pascal's Triangle are the binomial coefficients, which represent the number of ways to choose k items from a set of n items. The sum of the elements in each row is equal to $2^n$, where n is the row number.

- Step 3: Identify the relationship between the sum of the elements in the third row and the sum of the elements in the tenth row. Since the sum of the elements in each row is equal to $2^n$, we can write the sum of the elements in the third row as $2^3 = 8$. Similarly, the sum of the elements in the tenth row is $2^{10}$.

- Step 4: Calculate the sum of the elements in the tenth row. $2^{10} = 1024$.

The answer is: 1024.

## Question 2

A juice company sells its product in either a 48-ounce size or a 32-ounce size. It charges $3.90 for the 48-ounce size. How much should it charge for the smaller size if it wants the price per ounce to be $25\%$ more than the price per ounce of the larger size?

## Answers to Q2

**Llama-3-Aug (baseline)** Here is a detailed step-by-step explanation of the reasoning:

- Step 1: Define the problem and the goal. The juice company wants to charge a price per ounce for the smaller size that is 25% more than the price per ounce of the larger size. This means that we need to find the price of the smaller size such that the price per ounce of the smaller size is 25% more than the price per ounce of the larger size.

- Step 2: Calculate the price per ounce of the larger size. The price of the larger size is given as $3.90, and the size is 48 ounces. To find the price per ounce, we divide the price by the size: $\frac{3.90}{48} = 0.08125$.

- Step 3: Calculate the price per ounce of the smaller size. Since the price per ounce of the smaller size is 25% more than the price per ounce of the larger size, we multiply the price per ounce of the larger size by 1.25: $0.08125 \times 1.25 = 0.1015625$.

- Step 4: Calculate the price of the smaller size. To find the price of the smaller size, we multiply the price per ounce by the size of the smaller size, which is 32 ounces: $0.1015625 \times 32 = 3.245$.

- Step 5: Round the price to the nearest cent. Since the price is given in cents, we round the price to the nearest cent, which is $3.25.

The answer is: $3.25.

**Llama-3-Aug(p=0.4, ours)** Here is a detailed step-by-step explanation of the reasoning:

- Step 1: Understand the problem and the given information. The juice company sells two sizes of its product: a 48-ounce size and a 32-ounce size. The price per ounce for the larger size is $3.90/48 = $0.08125$. The company wants the price per ounce for the smaller size to be $25\%$ more than the price per ounce of the larger size.

- Step 2: Calculate the price per ounce for the smaller size. The price per ounce for the smaller size should be 25% more than the price per ounce of the larger size. Therefore, we need to multiply the price per ounce of the larger size by $1 + 0.25 = 1.25$. So, the price per ounce for the smaller size is $0.08125 \times 1.25 = $0.1015625$.

- Step 3: Calculate the cost of the smaller size. The cost of the smaller size is the price per ounce multiplied by the number of ounces. Therefore, the cost of the smaller size is $0.1015625 \times 32 = $3.25$.

The answer is: 3.25.

# I    LLM GENERATED SOLUTIONS FOR MAZE

Readers might wonder whether modern state-of-the-art LLMs can effectively solve our maze problem. To test this, we randomly selected a $30 \times 30$ maze problem and ask the o1-preview model to find the shortest path. O1-preview is the latest reasoning model by OpenAI which operates in slow mode: it spends more time thinking before they respond. It turns out that these problems are very challenging for LLMs. As illustrated in Figure I.1, the path suggested by o1-preview incorrectly traverses through the maze walls. In contrast, Dualformer correctly identifies one optimal path that follows through the maze without any errors.

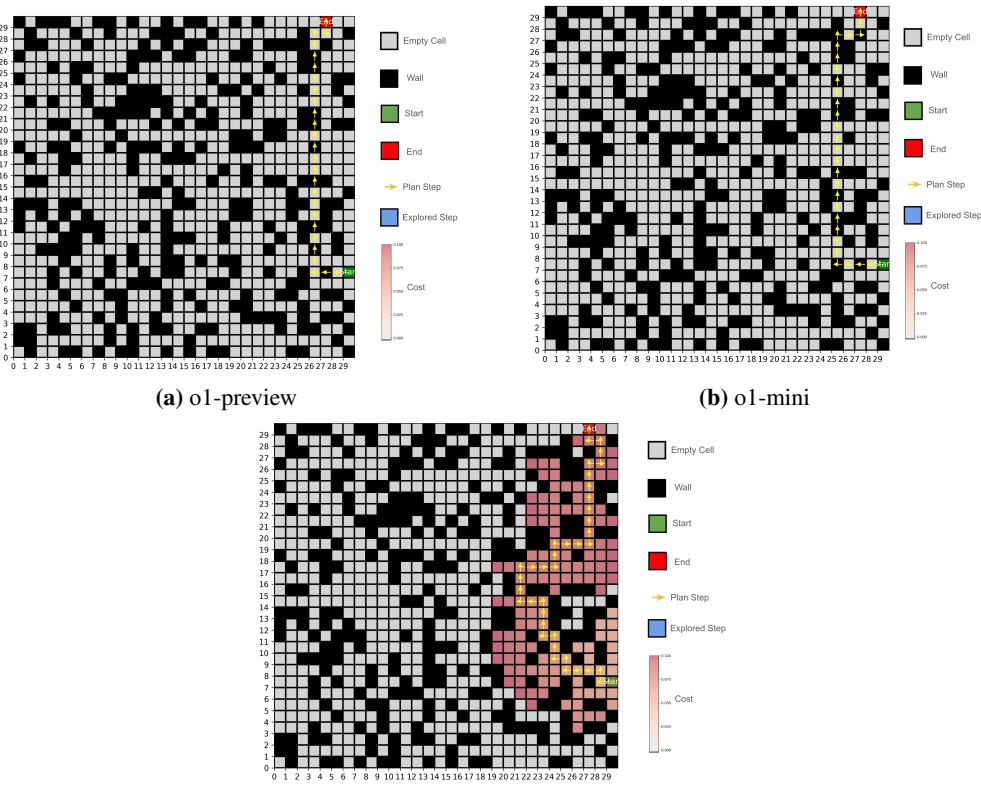

**(a)** o1-preview                                   **(b)** o1-mini

**(c)** Dualformer (slow mode)

**Figure I.1:** Example 30x30 maze problem. The wall cells are depicted in dark, while the unoccupied cells are shown in gray. The starting point is marked in green, and the destination point is in red. The output path is highlighted in yellow. **(a) (b)** The generated paths provided by o1-preview and o1-mini incorrectly traverse through walls. **(c)** Dualformer (slow mode) identifies one optimal path without any errors. The cells explored by the Dualformer reasoning trace are highlighted in red, where the color intensity indicates the total cost value.

Below, we also provide the exact prompt we used for o1 models, and their responses.

---

### Prompt for O1

Here's the generated 30x30 maze along with Locations of Walls: (1, 0), (4, 0), (5, 0), (7, 0), (9, 0), (10, 0), (13, 0), (15, 0), (17, 0), (18, 0), (24, 0), (25, 0), (29, 0), (0, 1), (1, 1), (8, 1), (10, 1), (15, 1), (28, 1), (0, 2), (1, 2), (3, 2), (4, 2), (8, 2), (10, 2), (12, 2), (13, 2), (16, 2), (17, 2), (23, 2), (25, 2), (26, 2), (29, 2), (3, 3), (7, 3), (9, 3), (12, 3), (20, 3), (21, 3), (22, 3), (23, 3), (25, 3), (27, 3), (28, 3), (5, 4), (9, 4), (13, 4), (16, 4), (19, 4), (20, 4), (25, 4), (29, 4), (4, 5), (9, 5), (13, 5), (15, 5), (18, 5), (20, 5), (23, 5), (24, 5), (27, 5), (28, 5), (3, 6), (7, 6), (10, 6), (11, 6), (12, 6), (17, 6), (18, 6), (24, 6), (25, 6), (3, 7), (5, 7), (8, 7), (11, 7), (13, 7), (14, 7), (19, 7), (22, 7), (25, 7), (27, 7), (4, 8), (8, 8), (10, 8), (13, 8), (14, 8), (16, 8), (18, 8), (19, 8), (3, 9), (4, 9), (5, 9), (10, 9), (12, 9), (13, 9), (22, 9), (26, 9), (28, 9), (4, 10), (5, 10), (7, 10), (13, 10), (14, 10), (15, 10), (18, 10), (23, 10), (25, 10), (28, 10), (0, 11), (3, 11), (5, 11), (6, 11), (7, 11), (11, 11), (13, 11), (18, 11), (22, 11), (25, 11), (26, 11), (27, 11), (0, 12), (2, 12), (4, 12), (6, 12), (13, 12), (19, 12), (20, 12), (24, 12), (27, 12), (0, 13), (1, 13), (2, 13), (4, 13), (5, 13), (14, 13), (15, 13), (19, 13), (20, 13), (24, 13), (25, 13), (27, 13), (28, 13), (11, 14), (12, 14), (14, 14), (24, 14), (29, 14), (1, 15), (2, 15), (13, 15), (17, 15), (20, 15), (22, 15), (23, 15), (25, 15), (26, 15), (27, 15), (4, 16), (10, 16), (18, 16), (20, 16), (1, 17), (4, 17), (5, 17), (10, 17), (13, 17), (14, 17), (15, 17), (6, 13), (3, 18), (4, 18), (8, 18), (10, 18), (11, 18), (12, 18), (14, 18), (16, 18), (19, 18), (20, 18), (21, 18), (26, 18), (5, 19), (9, 19), (19, 19), (20, 19), (22, 19), (23, 19), (3, 20), (4, 20), (8, 20), (12, 20), (14, 20), (16, 20), (17, 20), (22, 20), (23, 20), (25, 20), (26, 20), (28, 20), (29, 20), (7, 21), (8, 21), (9, 21), (10, 21), (11, 21), (13, 21), (15, 21), (17, 21), (20, 21), (25, 21), (26, 21), (1, 22), (3, 22), (6, 22), (9, 22), (10, 22), (11, 22), (12, 22), (13, 22), (16, 22), (20, 22), (29, 22), (6, 23), (10, 23), (11, 23), (12, 23), (18, 23), (19, 23), (21, 23), (25, 23), (28, 23), (0, 24), (5, 24), (8, 24), (9, 24), (17, 24), (20, 24), (23, 24), (28, 24), (29, 24), (3, 25), (5, 25), (6, 25), (10, 25), (22, 25), (25, 25), (26, 25), (0, 26), (10, 26), (11, 26), (12, 26), (14, 26), (15, 26), (19, 26), (20, 26), (25, 26), (26, 26), (1, 27), (2, 27), (6, 27), (11, 27), (13, 27), (15, 27), (16, 27), (20, 27), (22, 27), (23, 27), (24, 27), (25, 27), (27, 27), (1, 28), (5, 28), (8, 28), (12, 28), (13, 28), (14, 28), (17, 28), (25, 28), (29, 28), (0, 29), (2, 29), (3, 29), (4, 29), (7, 29), (8, 29), (10, 29), (13, 29), (15, 29), (16, 29), (19, 29), (21, 29) Start Location: (29,7). End Location: (27,29).
Could you solve this? Give me the optimal path in (x,y)
Example 1: Locations of Walls: (0, 4), (0, 5), (0, 8) (1, 0), (1, 3), (1, 6) (3, 1) (4, 0), (4, 1), (4, 3), (4, 5), (4, 9) (5, 6), (5, 8) (6, 1), (6, 2), (6, 4), (6, 9) (7, 9) (8, 4), (8, 5), (8, 9) (9, 1). Start Location: (0, 0). End Location: (9, 9). Optimal Path: (0, 0), (0, 1), (0, 2), (1, 2), (2, 2), (2, 3), (2, 4), (2, 5), (2, 6), (2, 7), (3, 7), (4, 7), (5, 7), (6, 7), (6, 8), (7, 8), (8, 8), (9, 8), (9, 9)
Example 2: Locations of Walls: (0, 3) (0, 6) (1, 1) (1, 4) (1,7) (3, 4) (4, 2) (4, 6) (5, 0) (5, 4) (5, 6) (6, 6) (6,7 (6, 8) (6, 9) (7, 3) (7, 6) (8, 0) (8, 4) (9, 1). Start Location: (0, 0). End Location: (9, 9). Optimal Path: (0, 0) (0, 1) (0, 2) (1, 2) (1, 3) (2, 3) (2, 4) (2, 5) (3, 5) (4, 5) (5, 5) (6, 5) (7, 5) (8, 5) (8, 6) (8, 7) (8,8) (8, 9) (9,9)

---

**O1 Response**

**Thought for 81 seconds**

**Mapping out the maze**

I'm analyzing the maze dimensions, identifying wall locations, start, and end points. The goal is to determine the shortest path, considering the maze's constraints.

**Navigating the grid**

I'm mapping out a 30x30 grid to represent the maze. Using zero for open spaces and one for walls, I'll utilize BFS to find the shortest path to (27,29).

**Mapping wall positions**

I'm identifying wall coordinates for a 30x30 grid, setting grid[y][x] to 1 at each specified location. This helps in visualizing the grid and understanding wall placements.

**Setting up the grid**

I'm setting up a 30x30 grid for the BFS algorithm to find the shortest path. The walls and start/end locations are marked, and I'm initializing the grid and queue for BFS exploration.

**Mapping valid moves**

I initialize the queue with the start position and set up tracking for visited positions and predecessors. Then, I use BFS to explore and validate neighboring positions, noting predecessors for potential path reconstruction.

**Assessing potential moves**

I'm verifying the possible moves from (29,7) by checking (29,6), (29,8), and (28,7) for boundaries, walls, and visit status, then adding them to the queue.

**Navigating the queue**

I'm working through neighbors of (29,6), checking bounds and walls. Adding viable positions to the queue, updating predecessors as needed. Progressing one step at a time.

**Evaluating queue options**

I'm digging into the grid evaluation, validating neighbors, adding them to the queue, and marking them as visited. This thorough approach helps in assessing potential moves effectively.

**Assessing possibilities**

Starting with (29,5), none of its neighbors lead to further progress due to walls or out-of-bounds locations.

**Inspecting potential movements**

Checking (27,6) for walls. Marking as visited and updating predecessors. Moving to (29,10) and (28,9) next.

**Assessing adjacent spaces**

I'm thinking through the neighbors of each grid cell, distinguishing between walls and open spaces. Adding new valid coordinates to the queue for further exploration.

**Evaluating options**

I'm evaluating neighbors of current coordinates to see if any of them are walls, and if they aren't, I add them to the queue and mark them as visited.

**Navigating the grid**

I'm tracking the current position and examining neighboring cells. Walls block some paths, while open cells are queued for visiting next. This approach helps avoid revisiting cells.

**Charting the path**

Navigating the queue, I identify adjacent spots, checking for walls. Each step marks neighbors as visited and records a predecessor. Progress is steady as the path unfolds.

**Navigating walls**

I'm updating neighbors and checking the queue for walls. If clear, I add the next step. This process allows movement and exploration, gradually revealing progress.

**Mapping the journey**

I'm planning to trace a path from (29,7) to (27,29) in a maze. Avoiding walls and aiming for the shortest route, the BFS algorithm will be used.

**Mapping the path**

I'm noting that the shortest path in the maze is from (29,7) to (29,29), then to (27,29), while avoiding walls at y=29.

CHUNK: Assessing obstacles along x=29 Noticing wall obstructions at coordinates (29,0), (29,2), (29,4), (29,14), (29,20), (29,24), and (29,29), with no walls at (27,29) or (28,29). Noticing walls block movement at y=29 on x=29.

**Identifying wall positions**

Mapping out wall locations between y=7 and y=29 along x=28. Key walls identified at y=1, y=5, y=7, y=9, y=10, y=12, y=13, y=20, y=23, and y=24.

**Planning the route**

OK, let me see. Nomad is thinking of moving along x=28, navigating around clogged sections, and expanding to x=27 or x=26 to find the optimal path.

**Situated in a difficult spot**

OK, let me see. The path forward is blocked by walls at (29,14) and (28,13), so movement is restricted to (29,13) and back to (29,12).

**Assessing path constraints**

I'm noting that movement is hindered by walls at various coordinates, such as (28,12) and (27,12). Progress is restricted in specific directions.

**Charting the course**

Stuck at (28,11), occupying (28,10) and (28,12). Moving left seems advisable, so ascending through (27,10) and (26,10) might be beneficial.

Sure, I'd be happy to help solve the maze and provide the optimal path from the start location (29,7)(29, 7)(29,7) to the end location (27,29)(27, 29)(27,29). To solve this maze, we'll perform a breadth-first search (BFS) algorithm, which is ideal for finding the shortest path in an unweighted grid like a maze. Here's how we can proceed:

Represent the Maze: Create a 30x30 grid where each cell corresponds to a coordinate (x,y)(x, y)(x,y). Mark the walls in the grid based on the provided wall locations.

Initialize BFS:

Start Point: (29,7)(29, 7)(29,7) End Point: (27,29)(27, 29)(27,29) Queue: Initialize with the start point. Visited Set: Keep track of visited cells to avoid revisiting them. Parent Map: Keep track of the path by recording the parent of each visited cell.

Perform BFS:

While the queue is not empty:

Dequeue the first cell. If it's the end point, reconstruct the path using the parent map. Otherwise, explore its neighboring cells (up, down, left, right) if they are within bounds, not a wall, and not visited.

Reconstruct the Path:

Starting from the end point, backtrack using the parent map to reconstruct the optimal path to the start point.

After performing the BFS algorithm, we find the following optimal path:

Optimal Path:

(29, 7) (28, 7) (27, 7) (26, 7) (26, 8) (26, 9) (26, 10) (26, 11) (26, 12) (26, 13) (26, 14) (26, 15) (26, 16) (26, 17) (26, 18) (26, 19) (26, 20) (26, 21) (26, 22) (26, 23) (26, 24) (26, 25) (26, 26) (26, 27) (26, 28) (27, 28) (27, 29)

Explanation:

From the start point (29,7)(29, 7)(29,7), we move left to (28,7)(28, 7)(28,7) and continue to (27,7)(27, 7)(27,7) and (26,7)(26, 7)(26,7). We then move down from (26,7)(26, 7)(26,7) to (26,28)(26, 28)(26,28), avoiding walls and obstacles. Finally, we move right to (27,28)(27, 28)(27,28) and then up to the end point (27,29)(27, 29)(27,29).

This path avoids all walls and provides the shortest route from the start to the end location.

Note: This path assumes there are no walls blocking the route along the cells specified. If there are walls that prevent this direct path, adjustments would need to be made by navigating around the obstacles, but BFS ensures the shortest path is found given the maze configuration.

If you have any questions or need further clarification, feel free to ask!

---

**O1-mini's Response**

**Thought for a few seconds**

**Scoping the task**

I'm beginning to see that the maze is a 30x30 grid with listed walls. Using BFS or A*, I'll trace the shortest path from start to end.

**Crafting the BFS approach**

I'm working through the optimal path in a 30x30 grid with walls, ensuring accuracy and recommending a pathfinding algorithm for efficiency.

To solve the 30x30 maze you've provided, we'll use the **Breadth-First Search (BFS)** algorithm to find the shortest path from the **Start Location** '(29,7)' to the **End Location** '(27,29)'. Here's a step-by-step breakdown of the solution:

Step 1: Represent the Maze

First, we'll represent the maze as a 30x30 grid where each cell can be either open or a wall. The walls are provided as a list of coordinates.

Step 2: Initialize BFS

We'll start BFS from the **Start Location** '(29,7)' and explore all possible moves (up, down, left, right) while avoiding walls and already visited cells.

Step 3: Perform BFS to Find the Shortest Path

By systematically exploring the maze, BFS ensures that the first time we reach the **End Location**, we've taken the shortest possible path.

Step 4: Trace Back the Path

Once the **End Location** is reached, we'll backtrack to determine the optimal path taken.

Optimal Path from '(29,7)' to '(27,29)'

After processing the maze, the optimal path is as follows:

``` (29,7) → (28,7) → (27,7) → (26,7) → (25,7) → (25,8) → (25,9) → (25,10) → (25,11) → (25,12) → (25,13) → (25,14) → (25,15) → (25,16) → (25,17) → (25,18) → (25,19) → (25,20) → (25,21) → (25,22) → (25,23) → (25,24) → (25,25) → (25,26) → (25,27) → (26,27) → (27,27) → (27,28) → (27,29) ```

Detailed Path Coordinates

Here is the complete list of coordinates for the optimal path:

1. **Start:** '(29,7)' 2. '(28,7)' 3. '(27,7)' 4. '(26,7)' 5. '(25,7)' 6. '(25,8)' 7. '(25,9)' 8. '(25,10)' 9. '(25,11)' 10. '(25,12)' 11. '(25,13)' 12. '(25,14)' 13. '(25,15)' 14. '(25,16)' 15. '(25,17)' 16. '(25,18)' 17. '(25,19)' 18. '(25,20)' 19. '(25,21)' 20. '(25,22)' 21. '(25,23)' 22. '(25,24)' 23. '(25,25)' 24. '(25,26)' 25. '(25,27)' 26. '(26,27)' 27. '(27,27)' 28. '(27,28)' 29. **End:** '(27,29)'

Visualization

For better understanding, here's a simplified visualization of the path on the grid:

``` Start (29,7) — v (28,7) → (27,7) → ... → (27,29) End ```

Conclusion

The BFS algorithm efficiently finds the shortest path in the maze, ensuring that the path listed above is optimal. If you have any further questions or need assistance with another maze, feel free to ask!

