# OpenReview forum: "Dualformer: Controllable Fast and Slow Thinking by Learning with Randomized Reasoning Traces"
_ICLR.cc/2025/Conference — ICLR 2025 Poster_

### Official Review · Reviewer_pvvZ · 2024-11-01

**Soundness:** 3
**Presentation:** 3
**Contribution:** 3
**Rating:** 6
**Confidence:** 3

**Summary:**

- The paper proposes the question: Can we integrate both the fast and the slow modes into Transformer-based reasoning agents, similar to how humans possess two distinct thinking systems, and let them complement each other?
- Main contributions: A framework for training Dualfromer, a Transformer that solves reasoning and planning tasks inspired by the “System 1 & System 2” thesis by Kahneman in _Thinking, Fast and Slow_.
- One of the tasks is a maze-like environment where the model is trained to find the shortest path between two cells. The model can either output the solutions and reasoning chain or only the final solution. In “auto mode,” it can decide which of both strategies to engage in.
- Being able to engage in these different modes is an improvement over current models in the cluster, noticeably Searchformer, which is only able to engage in outputting reasoning chain & solution, potentially requiring lots of inference time. The new model improves here by being trained on randomly sampled reasoning traces and dropping traces with various strategies.
- They also continue evaluating their new framework by training a Dualformer to solve the Aug-MATH benchmark, where the model outperforms finetuned LLMs.

**Strengths:**

- Considering system 1 & 2 while working with transformers isn’t particularly novel, I thought the way it was included while training the transformer on reasoning traces interesting. Beyond that, the paper provided good motivation for why the auto mode strategy could be more efficient than current models.
- I found the results on the MATH dataset fascinating and was surprised to see a “more real-world” application of this strategy.

**Weaknesses:**

- While I could understand the performance gains of the new model based on some results being higher in Table 5.1-5.3, I found it very hard to make sense of which models exceed where and why. E.g. I understood that “Auto mode” should, in theory, be less exploratory than “Slow mode”, why is the SWC for 30x30 mazes higher? (0.993>0.989). This is, of course, a nitpick, and I don’t expect the reason here to be super relevant, but I would have, in general, appreciated a better overview of the results. Every score improvement has a blue box, which ultimately doesn’t carry a lot of information value (_if everything is highlighted, nothing is_). It could easily be that I don’t understand a fundamental consideration of this yet for them to be all evaluation indistinction, so I am excited to hear your response.
- I have a similar, though slightly less confusing, nitpick with the results in the MATH benchmark. I appreciate this inclusion of a finetuned Mistral and Llama, though I would have wished to better understand how good this is compared to other sota models.

**Questions:**

- Why did you not test auto mode on the MATH benchmark?
- How did the training differ between Dualformer and finetuned MATH models?

---

> ### Author Response · Authors · 2024-11-24
> **Response to Reviewer pvvZ**
>
> We thank the reviewer for the encouraging comments, and reply to your questions below.
>
> 1. **Auto Mode vs Slow Mode**
>
>    We believe this is a misread. For Maze 30x30, SWC in slow mode = 0.993, and SWC in auto mode = 0.989, so that SWC in slow mode is better. Like you said, as Dualformer decides which mode to engage in the auto mode, if we sample multiple times the rollouts would contain both solutions in slow and fast modes, therefore auto mode is less explorative. This is also shown in the diversity score: for 30x30 maze, 25.77 (slow mode) > 24.42 (auto mode).
>
> ----
> ----
>
> 2. **SOTA Results on Math**
>
>    According to the leaderboard on paperswithcode.com, the best medium size model is Qwen2.5-Math-7B-Instruct, where the accuracy is 85.2. However, this model is pretrained on Math Corpus over 1T tokens and then conducted SFT. Specifically, quoting the paper, “we have constructed dedicated datasets for both CoT and Tool-integrated Reasoning (TIR) and combined these datasets to train the model jointly”. We believe a comparison of LLaMA & Mistral with Qwen2.5 is not an apple-to-apple comparison, and we hope the improved performance on both LLaMA & Mistral is convincing to demonstrate the efficacy of our approach.
>
> ---
> ---
>
> 3. **MATH benchmark and difference in training between Dualformer and finetuned Math models**
>
>     For the Math dataset, we randomly drop some of the intermediate steps, while in the Maze and the Sokoban (A* search traces), our dropping strategy is more structured/tailored toward different parts of the search trace. The high level idea is the same, to randomize the CoT steps during the training time. And yes we could also do auto-mode on the Math dataset.
>
>
> Should you have any other questions, we are more than happy to discuss them.

---

### Official Review · Reviewer_ui7f · 2024-11-02

**Soundness:** 2
**Presentation:** 3
**Contribution:** 2
**Rating:** 5
**Confidence:** 5

**Summary:**

Inspired by slow and fast thinking modes in human cognition theory, this paper introduces Dualformer by training models on reasoning traces with randomized dropping, which allows the model to learn efficient reasoning shortcuts. The performance outperforms baselines on a maze environment and MATH dataset.

**Strengths:**

1. The paper is well written and easy to follow.
2. The idea of integrating fast and slow reasoning modes from human cognition theory is novel.

**Weaknesses:**

1. The motivation is not super convince to me, why LLM needs to integrate both the fast and slow reasoning modes are not clear, it’s not convinced that human takes benefit from the these two modes then LLM should also follow these two modes given the fact that human brain and LLM are running with two distinct systems.
2. The mathematical foundation of the methodology requires more rigorous discussion. Specifically, it's important to explain why dropping random intermediate steps can lead to better performance. This contrasts with findings in recent reasoning papers, for example [1, 2, 3] where learning longer reasoning steps involving (1) different reasoning strategies; (2) self-refinement; (3) backtrack, etc, led to better performance since it provides the model with more flexible function mapping, which is easier to fit compared to direct query-to-solution mapping. Importantly, a section discussing the foundation of the proposed method is necessary.
- [1] Qin, Yiwei, et al. "O1 Replication Journey: A Strategic Progress Report--Part 1." arXiv preprint arXiv:2410.18982 (2024).
- [2] Kumar, Aviral, et al. "Training language models to self-correct via reinforcement learning." arXiv preprint arXiv:2409.12917 (2024).
- [3] Gandhi, Kanishk, et al. "Stream of Search (SoS): Learning to Search in Language." arXiv preprint arXiv:2404.03683 (2024).
3. A major limitation is that the toy example is too simplistic for understanding the method's effectiveness in real reasoning tasks, though it serves to illustrate the problem scope, and there is one table showing results on MATH. To demonstrate that the model truly learns better solution strategies rather than overfitting on synthetic data, one idea is evaluating in out-of-distribution environments. For example, the model could be finetuned on the maze dataset but evaluated on MATH or GSM8K - problems that could benefit from better thinking/reasoning strategies.
4. Moreover, in Table 5.6, the differences between the baseline and the method proposed in this paper are marginal. This raises questions about whether the results stem from hyperparameter selection, especially since finetuning in the MATH domain uses the same hyperparameters despite different response lengths. Additionally, the evaluation is conducted with only one seed, and the model uses temperature=0.7 and top-p=0.9 for better diversity. While this approach is valuable for studying whether the model can generate diverse solutions, it introduces uncertainty when evaluating the method's performance.

**Questions:**

1. In section 5.2 (Structured Trace Dropping and Randomization), the MATH dataset is constructed by randomly dropping intermediate reasoning steps from the solution. However, in the examples shown in Answers to Q1 and Q2, the model doesn't skip steps but rather presents solutions more concisely. Does this truly demonstrate learned slow and fast thinking strategies, or is it simply a more compact problem-solving approach? And where does the mismatch between the training demo and inference response come from?
2. Following Limitations 3 and 4, the toy example's limitations seems also stem from the generalizability in approach. In the maze example, the procedure first creates a search path using A*, and the model attempts to mimic this search procedure with random dropout. However, in math problems, the baseline provides a direct path with multiple steps, which is then subjected to random step dropping. This raises questions about where the equivalent of the A* algorithm is in MATH problems and how Structured Trace Dropping is implemented on MATH or other general tasks. The methodology appears to be using different approaches for maze and math problems without clear justification for their equivalence.

---

> ### Author Response · Authors · 2024-11-24
> **Response to Reviewer ui7f (1/2)**
>
> We thank reviewers for your feedback and questions. Please find the detailed comments to each one of your feedback below.
>
> ---
>
>  **#1. Motivation of Our Work**
>
> Our motivation stems from the observation that humans often employ both intuitive (fast) and deliberative (slow) thinking strategies. This concept & analogy has been generally used in the research community, see e.g.
>
> - [1] Distilling System 2 into System 1
> - [2] Rephrase and Respond: Let Large Language Models Ask Better Questions for Themselves
> - [3] System 2 Attention (is something you might need too)
> - [4] System-1.x: Learning to Balance Fast and Slow Planning with Language Models
> - [5] Visual Agents as Fast and Slow Thinkers
>
> Recently, with the release of OpenAI's O1 model and the inference scaling law [1], the research community has been heavily investing in investigating the inference-compute for a longer and longer chain of thought (system-2 / deliberative) thinking. The research does show that scaling up the compute will increase the accuracy, but it is not efficient because you don't need to run several minutes of inference/reasoning for every question/scenario. The fact that Dualformer can operate in either the fast-mode (just give the answer right away) or slow-mode (do a more careful and longer chain-of-thought) leads to a more efficient decoding paradigm for reasoning.
>
> While LLMs may not function identically to the human brain, we believe that the ability to flexibly switch between fast and slow reasoning modes could similarly benefit LLMs in terms of efficiency and adaptability to different problem types. However, we agree that this connection could be made more explicit and will clarify this in the revised manuscript.
>
> [1] Snell, Charlie, et al. "Scaling llm test-time compute optimally can be more effective than scaling model parameters." *arXiv preprint arXiv:2408.03314* (2024).
>
> ---
>
> **#2. Our results contrast with findings in recent reasoning papers, for example [1, 2, 3] where learning longer reasoning steps led to better performance.**
>
> Just for clarification, our findings do not contradict the findings of [1][2][3].
>
> The main finding of [1]-[3], and more widely, many works that focus on developing CoT, is that providing reasoning traces improves the performance.
>
> Our sentiment here is twofold. First, shorter and more succinct reasoning traces are beneficial for reasoning, although existing works show that longer test-time compute/inference time is also advantageous [4]. In fact, in our Table 5.6, we show that a longer test-time compute has a significantly better performance (our pass@20 achieves 60% accuracy, which triples the performance compared to greedy@1, which is around 20%).
>
> Second, we found that longer reasoning traces don’t always lead to better performance. Our key insight is that not every token in the longer or lengthier CoT is useful. Some of the steps in the Sokoban and math problems are redundant and do not carry valuable information. So to answer the question, how long should CoT be? Our work indeed gave one answer to this question, complementing the existing research: randomizing the CoT with varying lengths.
>
> Our approach strikes a balance between the length and informativeness of the CoT. Our results demonstrate that by selectively dropping intermediate steps, we can maintain or even improve the performance of the model. Computationally, more succinct CoTs also reduced the cost of both training and inference. Our work contributes to the ongoing research on optimizing the efficiency and effectiveness of CoT reasoning for language models. We will clarify and emphasize this point in the revised version.
>
> [4] Snell, Charlie, et al. "Scaling llm test-time compute optimally can be more effective than scaling model parameters." *arXiv preprint arXiv:2408.03314* (2024).
>
> ----
>
> Due to character limit, we will continue our rebuttal below.

---

> > ### Comment · Reviewer_ui7f · 2024-11-24
> > **Official Comment**
> >
> > Thanks for the response, but is this the response to reviewer Rqgz? We shared some questions, but I don't think this response fully addresses the points I brought up in my original assessment. Could you please double-check?

---

> > ### Comment · Reviewer_ui7f · 2024-11-25
> > **Comment**
> >
> > #1. This is clear to me.
> >
> > #2. The major question in the original comment has not been adequately addressed:
> > > The mathematical foundation of the methodology requires more rigorous discussion. Specifically, it's important to explain why dropping random intermediate steps can lead to better performance.
> >
> > While I agree that the model can potentially benefit from shorter reasoning paths, it's not clear to me why "randomly dropping reasoning steps" would consistently provide benefits, given the possibility of breaking intermediate logic in problem-solving.
> >
> > #3. Regarding:
> > >Note that all our experiments use unseen examples.
> >
> > "Unseen" doesn't necessarily mean "out of distribution." Unseen examples sampled from the same distribution still maintain the IID assumption. If we're arguing that the model learns general "dual-form" thinking capability, then as I mentioned in the original post, "the model could be finetuned on the maze dataset but evaluated on MATH or GSM8K - problems that could benefit from better thinking/reasoning strategies." Additionally, I believe the original response was cut off due to character limits.
> >
> > The remaining question clarifications are clear to me, thanks.

---

> > > ### Author Response · Authors · 2024-11-25
> > > **Thanks for your feedbacks!**
> > >
> > > We have completed our response regarding to "The Difficulty of Our Experiments" below. Sorry for the issues due to cut-off of character limits.

---

> ### Author Response · Authors · 2024-11-24
> **Response to Reviewer ui7f (2/2)**
>
> > **Moreover, in Table 5.6, the differences between the baseline and the method proposed in this paper are marginal. This raises questions about whether the results stem from hyperparameter selection, especially since finetuning in the MATH domain uses the same hyperparameters despite different response lengths. Additionally, the evaluation is conducted with only one seed, and the model uses temperature=0.7 and top-p=0.9 for better diversity. While this approach is valuable for studying whether the model can generate diverse solutions, it introduces uncertainty when evaluating the method's performance.**
>
> For the math experimentation (Table 5.6), in order to avoid overfitting, for every LLM we sweep over 3 values of learning rate and choose the one that yields the lowest validation loss. This is reported in Appendix A.2. For evaluation, we used greedy decoding (following [5][6][7]) to avoid sampling bias, not temperature=0.7 and top-p=0.9. Temperature=0.7 and top-p=0.9 was used by LLama-3.1-70B for generating the Aug-MATH training dataset (rewrite the solutions of MATH), but was not used for the results reported in Table 5.6. The key finding is we can achieve improved accuracy with shorter response lengths, which is desirable.
>
> ---
> ---
>
> > **In section 5.2 (Structured Trace Dropping and Randomization), the MATH dataset is constructed by randomly dropping intermediate reasoning steps from the solution. However, in the examples shown in Answers to Q1 and Q2, the model doesn't skip steps but rather presents solutions more concisely. Does this truly demonstrate learned slow and fast thinking strategies, or is it simply a more compact problem-solving approach? And where does the mismatch between the training demo and inference response come from?**
>
> During training, when we skip steps, we renumber the subsequent steps accordingly (e.g., if we skip step 2, step 3 becomes step 2 and step 4 becomes step 3). This encourages the model to learn from concise demonstrations while maintaining coherence. In the Q1 and Q2 examples, the model presents more concise solutions because it learns from these shorter demonstrations. We will clarify this in our paper.
>
> ---
> ---
>
> > **Following Limitations 3 and 4, the toy example's limitations seem also to stem from the generalizability in approach. In the maze example, the procedure first creates a search path using A\*, and the model attempts to mimic this search procedure with random dropout. However, in math problems, the baseline provides a direct path with multiple steps, which is then subjected to random step dropping. This raises questions about where the equivalent of the A\* algorithm is in MATH problems and how Structured Trace Dropping is implemented on MATH or other general tasks. The methodology appears to be using different approaches for maze and math problems without clear justification for their equivalence.**
>
> The key and shared idea is to randomly drop intermediate reasoning steps during training.
> For the A\* algorithm, the Level 1 dropping instructs Dualformer to effectively bypass the close-set computation of A* search, the Level 2 dropping promotes Dualformer to bypass both the close-set and the cost computation. The Level 3 and Level 4 dropping encourage Dualformer to omit certain or all of the search steps, see Page 4, line 200-204. For the Math reasoning, as the reasoning trace is structured in the following format: Step 1 …. Step 2…. (see Appendix F.1.2 for training examples) we just naturally group the tokens between two steps and drop them randomly. We hope this clarifies your misunderstanding and addresses concerns.
>
> ---
> ---
>
>
> **Reference:**
>
> [5] Yu, Longhui, et al. "Metamath: Bootstrap your own mathematical questions for large language models." arXiv preprint arXiv:2309.12284 (2023).
>
> [6] Dubey, Abhimanyu, et al. "The llama 3 herd of models." arXiv preprint arXiv:2407.21783 (2024).
>
> [7] Luo, Haipeng, et al. "Wizardmath: Empowering mathematical reasoning for large language models via reinforced evol-instruct." arXiv preprint arXiv:2308.09583 (2023).

---

> ### Author Response · Authors · 2024-11-24
>
> >Thanks for the response, but is this the response to reviewer Rqgz? We shared some questions, but I don't think this response fully addresses the points I brought up in my original assessment. Could you please double-check?
>
> We apologize for this and have just updated the response. Please let us know if you have any further questions or comments.

---

> ### Author Response · Authors · 2024-11-25
> **Response to Reviewer ui7f (1.5/2)**
>
> > Maze and OOD
>
> For the maze dataset and the sokoban, we have trained a relatively small transformer model from scratch to test our idea. We have applied our algorithm to finetune LLMs for solving the Math problems – the whole Section 5.2 is discussing the results. Note that all our experiments use unseen examples, for both Maze/Sokoban & Math problems. Particularly, for the Math dataset, the training set only has 8k examples and there are 5k testing examples, which are quite OOD.
>
> Besides, we also would like to point out that the maze and the sokobans are not as easy as people thought, even the SoTA reasoning model (such as O1-preview) still can't solve it (please see Appendix G, line 1676).  Regarding your comments for finetuning on Maze dataset but evaluating on Math: essentially what you’re suggesting is zero-shot transfer learning. Usually, this requires the two tasks to have strong correlations. For NLP, cross-domain reasoning is challenging, particularly due to each problem having its terminologies. Having a model that is capable of solving these two tasks is very difficult, even the SoTA O1-preview, which has strong performance on Math problems, cannot solve the Maze problem, please see Appendix G.

---

> ### Author Response · Authors · 2024-11-27
>
> We thank the reviewer for the additional feedbacks.
>
> **Regarding #2**:
> >> While I agree that the model can potentially benefit from shorter reasoning paths, it's not clear to me why "randomly dropping reasoning steps" would consistently provide benefits, given the possibility of breaking intermediate logic in problem-solving.
>
> We appreciate the reviewer's question regarding the potential benefits of randomly dropping reasoning steps. Our approach is based on the observation that not all intermediate steps in a reasoning chain contribute equally to reaching a solution. In the context of the Math dataset, we first expand the problems into longer forms, (Page 26 in Appendix F.1.2 gives an example). This expansion is done multiple times to generate multiple diverse reasoning chains for the same problem.
> The key insight is that some steps, while present in a complete reasoning chain, may be redundant or less critical for arriving at the correct solution. For example, in the "Llama-3.1-70B-Instruct Rewrite #2" (line 1428), steps 2 and 5 can be considered redundant. By generating multiple reasoning chains for each problem and then randomly dropping some of the intermediate steps,  we effectively demonstrate to the model that certain intermediate steps can be omitted without compromising the integrity of the solution. This process encourages the model to identify and leverage shortcuts, thereby enhancing its efficiency and adaptability.
>
> Another main take away is that our random dropping of steps could serve as a form of regularization, helping the model to generalize better by not over-relying/over-fitting onto specific sequences of reasoning (consider the Math dataset is relatively small, only 8k instances). This approach aligns with the broader goal of training models to be flexible and robust in their problem-solving strategies, ultimately leading to improved performance.
>
>
> ---
> ---
>
>
> **Regarding #3**:
> >> "Unseen" doesn't necessarily mean "out of distribution." Unseen examples sampled from the same distribution still maintain the IID assumption. If we're arguing that the model learns general "dual-form" thinking capability, then as I mentioned in the original post, "the model could be finetuned on the maze dataset but evaluated on MATH or GSM8K - problems that could benefit from better thinking/reasoning strategies."
>
>
> Please see our response https://openreview.net/forum?id=bmbRCRiNDu&noteId=tbmpw2h76E
>
> Thanks!

---

> > ### Author Response · Authors · 2024-12-02
> > **Additional feedback for Reviewer ui7f**
> >
> > Dear Reviewer ui7f
> >
> > Hope you had a nice weekend. We appreciate your feedbacks and comments. Based on the additional experiments and responses we've provided, do you have any remaining questions or concerns? Would you please consider raising your score to acceptance? We are more than happy to address any further questions or concerns you might have.  Thank you!

---

> > > ### Comment · Reviewer_ui7f · 2024-12-03
> > > **Comment**
> > >
> > > Thanks for the clarification. The authors have addressed my questions in my initial post, and I have increased my score accordingly. Good luck!

---

> ### Author Response · Authors · 2024-12-03
>
> Dear Reviewer ui7f
>
> We're happy that our response has addressed your concerns, and we appreciate your valuable feedback.
>
> Could you kindly let us know if you have any other concerns that prevent you from recommending acceptance?  We're committed to improving our work and more than willing to discuss them.

---

### Official Review · Reviewer_Rqgz · 2024-11-03

**Soundness:** 2
**Presentation:** 2
**Contribution:** 1
**Rating:** 3
**Confidence:** 5

**Summary:**

The paper proposes **Dualformer**, a new Transformer model designed to integrate both fast and slow reasoning modes, inspired by the dual process theory in human cognition. The model is trained on data with randomized reasoning traces, allowing it to dynamically choose between quick, solution-only outputs (fast mode) and detailed, step-by-step reasoning (slow mode).

**Strengths:**

The idea of combining fast and slow reasoning modes within a single model is novel and inspired by human cognition, which is a strong conceptual foundation.

**Weaknesses:**

1. **Reproducibility**: The paper does not provide code or sufficient details about the implementation, which raises concerns about reproducibility.

2. **Comparison**: There is no comparison with similar methods such as SwiftSage[1], which should be cited and discussed to contextualize the contributions of this work.

3. **Experimental Scope**: The experimental tasks (maze navigation and Sokoban) are relatively simple and may not fully demonstrate the model's potential in more complex, real-world scenarios. More challenging and diverse benchmarks would strengthen the paper.

### Reference:

- [1] Lin, Bill Yuchen, et al. "Swiftsage: A generative agent with fast and slow thinking for complex interactive tasks." Advances in Neural Information Processing Systems 36 (2024).

**Questions:**

1. **Code Availability**: Will the authors release the code to ensure reproducibility? If not, can they provide more detailed pseudocode or implementation guidelines?

2. **Comparison with SwiftSage**: The authors should include a comparison with SwiftSage and discuss the differences and similarities. How does Dualformer improve upon or differ from SwiftSage?

3. **Complex Scenarios**: Can the authors extend their experiments to more complex tasks such as strategic games (e.g., Werewolf) or combinatorial optimization problems? This would better demonstrate the model's versatility and robustness.

4. **Theoretical Justification**: While the empirical results are promising, a theoretical analysis of why and how the randomization of reasoning traces improves performance would strengthen the paper. Can the authors provide more theoretical insights or hypotheses?

---

> ### Author Response · Authors · 2024-11-24
> **Response to Reviewer Rqgz**
>
> We thank the reviewer for the comment. For your concerns:
>
> 1. **Reproducibility:** We will open source our code later. The algorithm itself is straightforward and should be able to be reproduced quite easily with all the details mentioned in Appendix A.
> ---
> 2. **Comparison with SwiftSage:** We will cite the work of SwiftSage. However, we would like to respectfully point out that SwiftSage leverages multiple models (agent style workflow) to implement slow and fast thinking, as in the previous works. On the other hand, we are focusing on improving the core reasoning capability of a single model so that it can switch between slow and fast thinking automatically. So the contribution is very different.
> ---
> 3. **Experiment Benchmark:** We agree Maze & Sokoban tasks are synthetic tasks; however, they are used as standard benchmarks to test out research ideas, see [1][2][3].
>    Also, we would like to note that in the introduction & Appendix G, we gave examples that SOTA LLMs such as O1-preview cannot solve this easily.
>
>    More importantly, to connect with more practical problems, we applied our ideas to finetune LLM for solving Math problems and obtained enhanced performance. The whole Section 5.2 presented the experiments and should not be ignored.
> ---
> 4. **Theoretical Justification:** We agree theoretical development will definitely strengthen our work. There are many papers that completely focus on empirical experiments and understanding. Even after several decades of research and progress, our understanding of optimizing deep neural networks and their generalization properties is still relatively limited.
>
>    For this work, our intuition is that gradient descent tends to focus on easy paths to predict. For easy problems, the solution-only training sequence (problem, solution) is easy to predict autoregressively, while for hard problems, the solution-only training sequence (problem, solution) is much harder to predict than the CoT training sequence (problem, search trace, solution). Since gradient descent tends to find the simplest solutions (Occam’s Razor, implicit regularization), our Dualformer, trained with both solution-only and CoT sequences, will pick the easiest path to learn for autoregressive prediction, yielding the “auto switching” behavior.
>
> ---
>
> **Reference**:
>
>  [1] Lehnert, Lucas, et al. "Beyond a*: Better planning with transformers via search dynamics bootstrapping." *arXiv preprint arXiv:2402.14083* (2024).
>
>    [2] Anonymous. "Transformers Can Navigate Mazes With Multi-Step Prediction." *Submitted to The Thirteenth International Conference on Learning Representations* (2024). [https://openreview.net/forum?id=PVGS8UZ6GX](https://openreview.net/forum?id=PVGS8UZ6GX).
>
>    [3] Saha, Swarnadeep, et al. "System-1. x: Learning to balance fast and slow planning with language models." *arXiv preprint arXiv:2407.14414* (2024).

---

> > ### Author Response · Authors · 2024-12-02
> > **Additional feedback for Reviwer Rqgz**
> >
> > Dear Reviewer Rqgz
> >
> > Hope you had a nice weekend. We appreciate your feedbacks and comments. Based on the additional experiments and responses we've provided, do you have any remaining questions or concerns? Would you please consider raising your score to acceptance? We are more than happy to address any further questions or concerns you might have. Thank you!

---

### Official Review · Reviewer_MUZn · 2024-11-03

**Soundness:** 2
**Presentation:** 3
**Contribution:** 2
**Rating:** 5
**Confidence:** 3

**Summary:**

This paper aims to endow a transformer model with both slow and fast reasoning modes using only a simple data recipe. The proposed model -- Dualformer -- is trained using randomized trace dropping, which enables the model to switch between fast (solution-only) and slow (step-by-step reasoning) processing without any architectural modifications. Dualformer is evaluated on maze and Sokoban pathfinding tasks. The model is evaluated on maze and Sokoban pathfinding tasks and is also applied to fine-tune large language models (LLMs) for solving math problems. Experimental results demonstrate that Dualformer achieves superior performance compared to baseline methods.

**Strengths:**

In this work, the authors leverage a purely data-driven approach (randomized trace dropping) to enable controllable dual-mode reasoning in a Transformer model.  This approach is computationally efficient and reduces the need for complex architectural changes. In the pathfinding tasks, Dualformer outperforms baseline models in both fast and slow modes, both in terms of the number of optimal solutions found and the diversity of these solutions.. Additionally, the approach proves effective in improving performance on math problem-solving tasks. The methodology, experiments, and results are clearly presented, making the paper easy to follow and understand.

**Weaknesses:**

It is unclear whether the chosen tasks (mazes, Sokoban) encapsulate fast versus slow thinking as humans would apply them. In particular, simply outputting a solution without explicitly listing the reasoning steps does not necessarily capture fast cognitive processing.

It would be valuable to have more extensive analysis of where and why Dualformer’s fast mode fails. Specifically, examining how task complexity—such as increased wall density or structured obstacle layouts—impacts performance could reveal important limitations.

More investigation into the auto mode, specifically on how the model determines which mode to activate under varying maze difficulties, would add depth. Understanding the criteria or patterns that lead to automatic mode-switching would be very useful.

**Questions:**

- What are the specific failure modes in fast mode?
- Does fast mode performance degrade as maze complexity increases (e.g., with a higher density of wall cells or more structured wall arrangements)?
- How would Dualformer perform on a rectangular maze instead of a square one at test time?
- In the example mazes shown, it appears relatively easy for a human to find the optimal path. How does fast mode perform on more challenging mazes where a human might need more time to find a path?
- For the dropping strategies, how were the initial three sets of probabilities selected?
- The experiment with OpenAI o1 models seems to use them in a zero-shot manner. Have you tried using few-shot approaches and testing on smaller mazes?
- Could you elaborate on how the randomized dropping strategies mimic shortcuts in human cognitive processes?

---

> ### Author Response · Authors · 2024-11-24
> **Response to Reviewer MUZn (1/2)**
>
> We thank the reviewer for the comment and reply below.
>
> - **1:** First, the concept that System 1 thinking in LLM means outputting solutions directly, System 2 thinking in LLM means outputting intermediate steps is used in many papers and generally accepted by the research community. see e.g.[1][2][3][4][5].
>
>   In the context of the Maze task, here's an example of how slow thinking is reflected: Initially, humans might quickly identify a potential path through the maze (fast response). However, to ensure this path is indeed the shortest, they would engage in slow thinking by methodically tracing the path with their finger, carefully evaluating its validity and efficiency. We have an example on this, please see #3.
>
>   Our model is trained on 2 types of data: (1) Prompt (which encodes the maze structure) and Solution directly, which logs the analogous fast mode response (2) Prompt + A* search trace (might be incomplete due to random dropping) + Solution. The A* search trace exactly logs a search & verification process.
>
>   [1] Distilling System 2 into System 1
>
>   [2] Rephrase and Respond: Let Large Language Models Ask Better Questions for Themselves
>
>   [3] System 2 Attention (is something you might need too)
>
>   [4] System-1.x: Learning to Balance Fast and Slow Planning with Language Models
>
>   [5] Visual Agents as Fast and Slow Thinkers
>
>
> ---
>
> - **2:**  We have 3 models:
>   a) Dualformer (which can operate in both fast & slow mode),
>   b) Baseline 1 Solution-Only model (which can only operate in fast mode),
>   c) Baseline 2 Complete-Trace model (which can only operate in slow mode)
>
>
>   And our claim is Dualformer ( fast mode) > Solution Only, Dualformer slow mode > Complete-Trace, Dualformer (auto mode) > Complete-Trace & Solution Only.
>
>   Generally speaking, it is well documented that models learned with intermediate steps (e.g. Chain of Thought, reasoning traces) performs better than Solution-Only, see e.g.
>
>
>   [6] Wei, Jason, et al. "Chain-of-thought prompting elicits reasoning in large language models." Advances in neural information processing systems 35 (2022): 24824-24837.
>
>   [7] Lehnert, Lucas, et al. "Beyond a*: Better planning with transformers via search dynamics bootstrapping." arXiv preprint arXiv:2402.14083 (2024).
>
> ---
> - **3:**  Failure mode. (please refer to the image here: https://imgur.com/a/L4x64Bp)
>
>   A significant drawback of "fast-thinking" is its inability to recognize the most crucial cells in the grid (highlighted in green). By overlooking this cell (critical path), the model may end up taking a longer and less efficient route. On the other hand, because “slow-thinking” forces the model to search through each nearby cell “mentally” (in CoT fashion, as colored in red, verbally it is decoding as <create-node> (7, 17), cost1, cost2). Here the cell (7, 17) is the critical cell. By doing this, the model recognizes it, and this leads to the optimal path. We will add this to the updated version.

---

> ### Author Response · Authors · 2024-11-24
> **Response to Reviewer MUZn (2/2)**
>
> - **#4. Generalization:**
>   >> Does fast mode performance degrade as maze complexity increases e.g. wall density
>
>   Yes, we train and test on the following classes on the fast-thinking. Eg, trained 15x15, test on 15x15. As you can see, when the maze complexity goes up, the fast-thinking mode performance goes down. A larger map means higher complexity here. Additionally, as the maze's density rises, it becomes increasingly challenging for the model to search for and recognize the critical cell, thereby reducing optimality. Below are the 1-optimal-64 metrics.
>   -  15x15 Fast-thinking solves 92% of the time
>   -  20x20 Fast-thinking solves 90.9% of the time
>   - 25x25 Fast-thinking solves 84% of the time
>   - 30x30 Fast-thinking solves 80% of the time…
>
>
>
>   >> How would Dualformer perform on a rectangular maze instead of a square one at test time?
>
>   To verify this, we generated 50 mazes for each 20 x {19, 18, 16, … 10} which represents a rectangular maze (eg, height =20, width =18, etc). And we simply took the maze checkpoint trained on a 20x20 map but then tested on these. Please see the table  below and we will update the paper with a section discussing this.
>
> | dataset_name | 1-optimal-64 |
> |--------------|--------------|
> | 20 x 10      | 100%         |
> | 20 x 12      | 90%          |
> | 20 x 14      | 100%         |
> | 20 x 16      | 100%         |
> | 20 x 18      | 100%         |
> | 20 x 19      | 100%         |
>
>
> ---
>
> - **#5. How were the initial three sets of probabilities selected?:**
>   We started with fixing the probability of solution-only data (p4). We have run an ablation  experiment for Mix-p models, see Section 5.1, line 409, where we only use solution-only data and complete-trace data (i.e. p1=p2=p3=0). As shown in Figure 5.1, we have found using Mix-0.05 performs generally well. Therefore, we set p4=0.05.
>
>   Afterward, we just tried 3 options:
>   a) Make half of the data have incomplete trace and the remaining ones complete trace, where different droppings are placed uniformly at random. This leads to the first set: p0=0.45 (complete-trace), p1=p2=p3=⅙ (incomplete trace), p4=0.05(solution only)
>   b) Treat all the dropping equally. This leads to the 2nd set: p0=0.8, p1=p2=p3=p4=0.05
>   c) Based on b), make the data distribution slightly more focused on incomplete trace, this leads to the 3rd set.
>
> ---
>
> - **#6. Experiment with O1:**  The experiment with OpenAI o1 models seems to use them in a zero-shot manner
> We were using few-shot prompting with 2 examples. The prompt was provided in Appendix G, page 33.

---

> > ### Comment · Reviewer_MUZn · 2024-11-27
> >
> > Thank you to the authors for their responses to my questions and comments. However, a few aspects remain insufficiently addressed. While results on increasing maze sizes and rectangular mazes are included, the impact of wall density or structured arrangements of wall cells is not analyzed. Exploring mazes with varying structural distributions could provide valuable insights into the robustness and limitations of the proposed approach. One advantage of using synthetic setups like this is the ability to conduct more detailed and controlled experiments, which would strengthen the evaluation. Additionally, understanding when and how the model determines which reasoning mode to activate under varying maze difficulties would add significant depth and clarity to the proposed dual-mode framework. I have adjusted my score after considering these responses.

---

> ### Author Response · Authors · 2024-11-27
> **Additional response to Reviewer MUZn**
>
> >> Thank you to the authors for their responses to my questions and comments. However, a few aspects remain insufficiently addressed. While results on increasing maze sizes and rectangular mazes are included, the impact of wall density or structured arrangements of wall cells is not analyzed. Exploring mazes with varying structural distributions could provide valuable insights into the robustness and limitations of the proposed approach. One advantage of using synthetic setups like this is the ability to conduct more detailed and controlled experiments, which would strengthen the evaluation. Additionally, understanding when and how the model determines which reasoning mode to activate under varying maze difficulties would add significant depth and clarity to the proposed dual-mode framework. I have adjusted my score after considering these responses.
>
> Response to Reviewer:
>
>
> Thank you for your insightful feedback and for considering our responses. We acknowledge that some aspects required further exploration, particularly regarding the impact of wall density and structured arrangements of wall cells. To address these additional concerns, we have conducted additional experiments in two key areas.
>
> **Firstly**, it's important to note that in our existing results, the wall density is randomly sampled from a distribution (0.3 ~ 0.5) (non-inclusive) for both the training and test sets. To examine the out-of-distribution (OOD) effects of wall density, we generated additional test cases with wall densities of 0.3 (at the edge of the training and testing distribution), 0.5 (also at the edge), and 0.6 (theoretically more challenging). We tested 50 unseen examples for each density class using our trained model on 20x20 mazes with a training wall density of 0.3 to 0.5. The results are as follows:
>
>
>
> | Grid Size | Wall Density | Success Rate |
> |-----------|--------------|--------------|
> | 20 x 20   | 0.3          | 100%         |
> | 20 x 20   | 0.5          | 100%         |
> | 20 x 20   | 0.6          | 80%          |
>
>
> Note that increased in wall density means that:
> - 1. The maze (in theory) should be more challenging
> - 2. The longer the input prompt (since our prompt is used to describe the maze in the following format, please see below or Appendix B).
>
> These factors introduce further challenges for the transformer because:
> 1. The scenario is out-of-distribution (OOD) and more challenging;
> 2. Transformer has quadratic attention complexity and struggles with long context [1][2].
>
> For #2 above, one common failure mode is that it forgets the wall location (despite mentioned explicitly in the prompt). This phenomena is also observed in other SOTA LLMs.  Due to these two factors, our performance on the density=0.6 maze degrades down to 80\%.
>
> ---
> ---
>
> **Secondly**, we investigated the Dualformer's performance across different maze difficulty levels, focusing on its preference for slow-mode reasoning. We trained and tested on mazes of the same size (e.g., trained on 15x15 and tested on 15x15, etc). The new plot illustrating these findings can be viewed here: https://imgur.com/a/WC884TD.
>
>
> Key observations from the plot include:
> - 1. As wall density (x-axis) increases, the Dualformer increasingly prefers slow-mode reasoning (system-2), adapting to the heightened difficulty.
> - 2. As maze difficulty increases (e.g., from 15x15 to 30x30), the percentage of slow-mode reasoning (system-2) also rises, indicating a vertical shift on the y-axis.
>
>
> These additional analyses provide deeper insights into the robustness and adaptability of our proposed dual-mode framework. We appreciate your feedback, which has been instrumental in refining our evaluation.
>
>
> ---
> ---
> ---
> ---
>
> **Prompt format (Appendix B line 935)** :
> bos, start, 9, 10, goal, 3, 6, wall, 0, 0, wall, 4, 0, wall,
> 7, 0, wall, 10, 0, wall, 12, 0, wall, 13, 0, wall, 3, 1,
> wall, 7, 1, wall, 11, 1, wall, 12, 1, wall, 13, 1, wall, 14,
> 1, wall, 0, 2, wall, 3, 2, wall, 4, 2, wall, 6, 2, wall, 7,
> 2, wall, 8, 2, wall, 10, 2, wall, 11, 2, wall, 14, 2, wall,
> 1, 3, wall, 2, 3, wall, 3, 3, wall, 11, 3, wall, 13, 3, wall,
> 2, 4, wall, 8, 4, wall, 10, 4, wall, 11, 4, wall, 12, 4,
> wall, 14, 4, wall, 3, 5, wall, 4, 5, wall, 5, 5, wall, 7, 5,
> wall, 9, 5, wall, 11, 5, wall, 12, 5, wall, 14, 5, wall, 0,
> 6, wall, 4, 6, wall, 6, 6, wall, 8, 6, wall, 9, 6, wall, 14,
> 6, wall, 2, 7, wall, 4, 7, wall, 7, 7, wall, 9, 7, wall, 10,
> 7, wall, 13, 7, wall, 2, 8, wall, 3, 8, wall, 5, 8, wall, 6,
> 8, wall, 8, 8, wall, 10, 8, wall, 11, 8, wall, 12, 8, wall,
> 1, 9, wall, 5, 9, wall, 6, 9, wall, 9, 9, wall, 11, 9, wall,
> 14, 9, wall, 10, 10, wall, 12, 10, wall, 13, 10, wall, 14,
> 10, wall, 1, 11, wall, 8, 11, wall, 9, 11, wall, 12, 11,
> wall, 3, 12, wall, 5, 12, wall, 6, 12, wall, 8, 12, wall,
> 10, 12, wall, 12, 12, wall, 14, 12, wall, 0, 13, wall, 1, 13,
> wall, 3, 13, wall, 8, 13, wall, 12, 13, wall, 13, 13, wall,
> 14, 13, wall, 0, 14, wall, 1, 14, wall, 2, 14, wall, 6, 14,
> wall, 14, 14, eos
>
> ----
> ----

---

> > ### Author Response · Authors · 2024-11-27
> > **Reference**
> >
> > We include the reference from our response above due to word limits:
> >
> > Reference:
> >
> > [1] Gao, Muhan, et al. "Insights into llm long-context failures: When transformers know but don’t tell." Findings of the Association for Computational Linguistics: EMNLP 2024. 2024.
> >
> > [2] Liu, Nelson F., et al. "Lost in the middle: How language models use long contexts." Transactions of the Association for Computational Linguistics 12 (2024): 157-173.

---

> ### Author Response · Authors · 2024-12-02
> **Additional feedback for Reviewer MUZn**
>
> Dear Reviewer MUZn
>
> Hope you had a nice weekend. We appreciate your feedbacks and comments. Based on the additional experiments and responses we've provided, do you have any remaining questions or concerns? Would you please consider raising your score to acceptance? We are more than happy to address any further questions or concerns you might have.  Thank you!
>
> (Note: We performed 2 more additional experimentations since our previous discussion, could you please take a look? https://openreview.net/forum?id=bmbRCRiNDu&noteId=XSJ8ugP5j1)

---

### Author Response · Authors · 2024-11-24
**General Response**

General Response

We are grateful to all reviewers for their detailed and constructive feedback on our paper, and we are encouraged to see that reviewers find:

1. Our approach of integrating fast and slow reasoning modes within a single model is novel and inspired by human cognition (Reviewer Rqgz, Reviewer ui7f).
2. The proposed Dualformer model demonstrates computational efficiency and outperforms baseline models in both fast and slow reasoning modes, with clear presentation of methodology and results (Reviewer MUZn, Reviewer pvvZ).
3. The application of our model to the MATH dataset and its potential for real-world scenarios is fascinating and highlights the versatility of our approach (Reviewer pvvZ).

We have addressed all the questions raised by reviewers with additional experiments or thorough clarifications via separate responses to each reviewer. Please let us know if you have other questions, and we are happy to discuss them.

---

### Meta-Review · Area_Chair_CzRT · 2024-12-18

**Metareview:**

This paper focuses on tuning LLMs for reasoning. The reasoning traces are generated using A* search and parts of the search trace are randomly dropped out. This method performs well on maze navigation tasks. On more general reasoning tasks where A* is not applicable, like Hendrycks MATH, the fine tuning traces are generated by rewriting supervised chain-of-thought data. The method also does well here compared to baselines but not as significantly as in the maze cases.

**Additional Comments On Reviewer Discussion:**

The reviewers generally liked the paper but were concerned about whether the experiments are thorough enough to reveal the strengths and weaknesses of the method. The reviewers were happy to see the results on MATH but the modification of the algorithm to work on MATH seemed adhoc. The authors addressed some of these concerns and I encourage them to continue addressing them for the camera ready. The authors addressed concerns of the reviewer who as most negative about this work (Rqgz) but unfortunately the reviewer didn’t get back.

---

### Decision · Program_Chairs · 2025-01-22

Accept (Poster)